



# The nitrogen budget of laboratory-simulated western U.S.
# wildfires during the FIREX 2016 FireLab study
James M. Roberts[1], Chelsea E. Stockwell[1,2@], Robert J. Yokelson[3], Joost de Gouw[1,2,5], Yong Liu[4],
Vanessa Selimovic[3], Abigail R. Koss[1,2,5*], Kanako Sekimoto[1,2,6], Matthew M. Coggon[1,2], Bin
Yuan[1,2,†], Kyle J. Zarzana[1,2, Δ], Steven S. Brown[1], Cristina Santin[7], Stefan H. Doerr[7], and Carsten
Warneke[1,2]
[1]NOAA Earth System Research Laboratories (ESRL), Chemical Sciences Laboratory, Boulder, CO, USA.
[2]Cooperative Institute for Research in Environmental Sciences, University of Colorado Boulder, Boulder, CO, USA.
[3]Department of Chemistry and Biochemistry, University of Montana, Missoula, MT, USA.
[4]Department of Chemistry, University of Colorado, Denver, Denver, Colorado, USA.
[5]Department of Chemistry, University of Colorado Boulder, Boulder, CO, USA.
[6]Graduate School of Nanobioscience, Yokohama City University, Yokohama, Japan.
[7]Department of Geography, Swansea University, Swansea, UK.
* now at Tofwerk, USA, Boulder, CO, USA.
† now at Institute for Environmental and Climate Research, Jinan University, Guangzhou, China.
@ now at Scientific Aviation, Boulder, CO., USA.
Δ now at Department of Chemistry, University of Colorado Boulder, Boulder, CO, USA.
*Correspondence to:* James M. Roberts (james.m.roberts@noaa.gov)
**Abstract.** Total reactive nitrogen ($N_r$, defined as all nitrogen-containing compounds except for $N_2$ and $N_2O$) was
measured by catalytic conversion to NO and detection by $NO-O_3$ chemiluminescence together with individual $N_r$
species during a series of laboratory fires of fuels characteristic of Western U.S. wildfires, conducted as part of the
FIREX FireLab 2016 study. Data from 75 stack fires were analyzed to examine the systematics of nitrogen emissions.
The $N_r$/total-carbon ratios measured in the emissions were compared with fuel and ash N/C ratios and mass to estimate
that a mean (±std. dev.) of 0.68 (±0.14) of fuel nitrogen was emitted as $N_2$ and $N_2O$. The remaining fraction of $N_r$ was
emitted as individual compounds: nitric oxide (NO), nitrogen dioxide ($NO_2$), nitrous acid (HONO), isocyanic acid
(HNCO), hydrogen cyanide (HCN), ammonia ($NH_3$), and 44 nitrogen-containing volatile organic compounds
(NVOCs). The relative difference between the total reactive nitrogen measurement, $N_r$, and the sum of measured
individual $N_r$ compounds had a mean (±std. dev) of 0.152 (±0.098). Much of this "unaccounted" $N_r$ is expected to be
particle-bound species, not included in this analysis.
A number of key species, e.g. HNCO, HCN and HONO, were confirmed not to correlate only with flaming
or only with smoldering combustion when using modified combustion efficiency (MCE = $CO_2/(CO + CO_2)$) as a
rough indicator. However, the systematic variations of the abundance of these species relative to other nitrogen-
containing species were successfully modeled using positive matrix factorization (PMF). Three distinct factors were
found for the emissions from combined coniferous fuels, aligning with our understanding of combustion chemistry in
different temperature ranges: a combustion factor (Comb-N) (800-1200°C) with emissions of the inorganic
compounds NO, $NO_2$ and HONO, and a minor contribution from organic nitro compounds ($R-NO_2$); a high-
temperature pyrolysis factor (HT-N) (500-800°C) with emissions of HNCO, HCN and nitriles; and a low-temperature
pyrolysis factor (LT-N) (<500°C) with mostly ammonia, and NVOCs, with the temperature ranges being based on
known combustion and pyrolysis chemistry considerations. The mix of emissions in the PMF factors from the




chaparral fuels had a slightly different composition: the Comb-N factor was also mostly NO, with small amounts of
HNCO, HONO and $NH_3$, the HT-N factor was dominated by $NO_2$ and had HONO, HCN, and HNCO, and the LT-N
factor was mostly $NH_3$ with a slight amount of NO contributing. In both cases, the Comb-N factor correlated best with
$CO_2$ emission, while the HT-N factors from coniferous fuels correlated closely with the high temperature VOC factors
recently reported by Sekimoto et al., (2018) and the LT-N had some correspondence to the LT-VOC factors. As a
consequence, $CO_2$ is recommended as a marker for combustion $N_r$ emissions, HCN is recommended as a marker for
HT-N emissions and the family $NH_3$/particle ammonium is recommended as a marker for LT-N emissions.

## 1 Introduction

Wildfires have severe impacts on the chemistry of the atmosphere from local to global scales (Crutzen and

Andreae, 1990). A warmer, drier climate in western North America, coupled with policies that have allowed build-up
of fuels in forest ecosystems has led to increases in frequency and severity of wildfires in this region (Abatzoglou and
Williams, 2016; Westerling et al., 2006). The new strategy for management of wildfire in the U.S. is to allow fire
where possible and to fight fire where needed (Lee et al., 2014). The science behind making these decisions and
understanding their consequences involves, in part, a better understanding of the emissions from wildfires. The NOAA
FIREX (Fire Influence on Regional and Global Environments Experiment) FireLab experiment was conducted in the
Fall of 2016, at the U.S. Forest Service Fire Sciences Laboratory in Missoula, Montana, to acquire detailed
measurements of particle and gas-phase emissions from fires involving fuels characteristic of the western U.S.
(NOAA, 2018). Several aspects of these measurements dealing with VOC species, and individual reactive nitrogen
species ($N_r$, defined as all nitrogen compounds except for $N_2$ and $N_2O$) have already been published (Koss et al., 2018;
Manfred et al., 2018; Sekimoto et al., 2018; Selimovic et al., 2018; Zarzana et al., 2018), including emissions factors
for many of the $N_r$-species (Koss et al., 2018).

The $N_r$ compounds emitted by natural-convection biomass burning (BB) arise solely from the N in the fuels,

since the combustion temperatures are not high enough (<1200°C) to produce NOx from $N_2$ and $O_2$ (the so-called
Zeldovich or thermal nitrogen cycle) (Lobert and Warnatz, 1993; Taylor et al., 2004; Wotton et al., 2012). The fuel
nitrogen cycles that pertain to BB flaming combustion are shown schematically in Figure 1 (Glarborg et al., 2018;
Lobert and Warnatz, 1993; Lucassen et al., 2012). $N_r$ compounds are emitted as small molecules, HCN, HNCO and
$NH_3$ resulting from pyrolysis of the fuel, with minor contributions from larger N-containing organic species, especially
at lower temperatures. Flame chemistry converts those species to $N_2$, $N_2O$, NO, $NO_2$, and HONO as a result of radical
chemistry. It has been recognized for some time that a significant amount of denitrification (conversion of $N_r$
compounds to $N_2$) occurs due to reactions of NO with $NH_i$ (where i= 1, 2, or 3) or N atoms, as confirmed
experimentally (Kuhlbusch et al., 1991). While N atoms are also intermediates in the thermal $NO_x$ cycle and the
reaction $N + O_2 \Rightarrow NO + O$ figures in to both the fuel and thermal $NO_x$ cycles, the second reaction of the thermal $NO_x$
cycle, $O + N_2 = NO + N$, is too slow at BB flame temperatures to result in $NO_x$ production (Manion et al., 2015). In
addition to the small molecules shown in Figure 1, numerous $N_r$-compounds are emitted in roughly the following
categories: amides, amines, heterocyclic compounds, nitriles, isocyanates, and nitro compounds (Andreae, 2019;
Andreae and Merlet, 2001; Koss et al., 2018; Lobert et al., 1991; Lobert et al., 1990; Lobert and Warnatz, 1993;



81 Stockwell et al., 2015). These compounds are produced at much lower abundance from fuel pyrolysis and partial

82 reactions with the radical species in Figure 1.

83  The emissions of N-compounds from BB and wildfires in general have been the subject of considerable

84 research (Akagi et al., 2011; Andreae, 2019; Andreae and Merlet, 2001; Burling et al., 2010; Coggon et al., 2016;

85 Gilman et al., 2015; Kuhlbusch et al., 1991; Lobert et al., 1991; Lobert et al., 1990; Lobert and Warnatz, 1993;

86 McMeeking et al., 2009; Stockwell et al., 2015; Veres et al., 2010; Warneke et al., 2011; Yokelson et al., 2013b;

87 Yokelson et al., 2009). The known N-compounds range in oxidation state from $NH_3$ to $HNO_3$ and include $N_2$ and $N_2O$.

88 Among the more prominent and important $N_r$ species are: $NO_x$ (NO and $NO_2$) which is a key player in the atmospheric

89 oxidant cycle; $NH_3$ which has a major role in particle formation; nitrous acid (HONO) which can be an important

90 radical source; hydrogen cyanide and acetonitrile (HCN, $CH_3CN$) which are toxic at high concentrations and represent

91 valuable tracers for following fire transport; and isocyanates, isocyanic acid and methyl isocyanate (HNCO, $CH_3NCO$)

92 which have unique health impacts (Roberts et al., 2011). In addition, nitro (-$NO_2$), or nitrogen heterocyclic compounds

93 may contribute to so-called brown carbon, aerosol organic compounds exhibiting optical absorption in the near-UV

94 or blue wavelength regions. Wildfire N emissions also have very minor contributions from gas phase nitric acid

95 ($HNO_3$). Nitric acid is either not efficiently produced by BB or is readily incorporated into aerosol if it is produced in

96 fresh wildfire plumes, as is clear from the absence of $HNO_3$ enhancements in several studies of BB plumes (Liu et al.,

97 2016; Yokelson et al., 2009) (Alvarado et al., 2010), however nitrate ($NO_3^-$) has been shown to contribute to aerosol

98 mass particularly for inefficient combustion (May et al., 2014). Flame chemistry is inefficient in forming $N_2O$, relative

99 to the pathways that form $N_2$ (Andreae, 2019; Andreae and Merlet, 2001; Griffith et al., 1991; Hao et al., 1991). The

100 modeling of the emissions of these N-compounds on a large scale could benefit from a better understanding of the

101 total budget of these species as a function of fuel nitrogen content and the dependence of the individual species on

102 fuel type and combustion conditions.

103  The construction of $N_r$-budgets in this work is made possible by the inclusion of a total reactive nitrogen

104 measurement (termed $N_r$ herein), a method by which all nitrogen compounds besides $N_2$ and $N_2O$ are converted to NO

105 and detected by NO-$O_3$ chemiluminescence. This technology has been developed by a number of groups, typically

106 using precious metal or NiCr catalysts that have been shown to convert all $N_r$ compounds to NO (and to some extent

107 $NO_2$) at high temperatures (750-825°C) (Hardy and Knarr, 1982; Kashihira et al., 1982; Marx et al., 2012; Roberts et

108 al., 1988). There are also commercial instruments that incorporate this technology (see for example Thermo Scientific

109 Model 17i). This technique has been applied to gas phase atmospheric measurements, principally to measure $NH_3$ by

110 difference techniques (Saylor et al., 2010; Schwab et al., 2007), and has also been used to observe wildfire plumes

111 that have impacted ambient air measurements (Benedict et al., 2017; Prenni et al., 2014). We have recently developed

112 a platinum/molybdenum oxide $N_r$ catalyst system, and confirmed that it quantitatively converts $N_r$ compounds

113 including all particle-bound nitrogen compounds (Stockwell et al., 2018). To our knowledge this technique has not

114 been applied directly to BB emissions before.

115  This paper will describe the total reactive nitrogen, and individual $N_r$ compound measurements made during

116 the FireLab 2016 experiment. The total $N_r$ measurements will be combined with $CO_2$, CO, and VOC measurements

117 and fuel, residue and ash C and N content to estimate the amount of N lost to $N_2$ and $N_2O$. Fire-integrated $N_r$ will be





compared to fire-integrated measurements of individual compounds to determine the fraction of unaccounted-for $N_r$.
The systematic behavior of individual $N_r$ species and their fractional contribution to $N_r$ will be examined with respect
to fuel type, N content, and combustion processes. A positive matrix factorization (PMF) technique will be used to
examine commonalities between fires of different fuels under different conditions and compared to the PMF analysis
of the VOC emissions published by Sekimoto et al., (2018). The results will be used to arrive at suggested guidelines
that can be used estimate $N_r$-emissions profiles for fires representative of western North America.

**2 Methodology**

The FireLab 2016 study involved laboratory burns of fuels mostly characteristic of western North American

wildfires, but also some that have global significance such as Indonesian peat and yak dung. The procedures and
associated details of the study have been described previously by Selimovic et al., (2018) and will be only briefly
summarized here. The detailed data on fuel types, amounts and composition can be found in Table S1, and in the
Supplemental section of Selimovic, et al., (2018). The laboratory burns involved fuel samples, ranging in mass from
0.26 to 6.02 kg. Fires were started without the addition of any contaminants, using an electric igniter (a series of NiCr
heating elements that were flash-heated electrically), and typically lasted from approximately 5 to 30 minutes.
Seventy-five fires were conducted in the configuration where the smoke was directed up the central stack of the facility
where it could be sampled simultaneously by all the instruments that measured gas phase species, and some of the
particle phase measurements. The sampling platform was about 15 m above the fire and the sampling took place in
well-mixed smoke approximately 5s after emission (Christian et al., 2004). Thirty-one additional fires were conducted
on most of the same fuels, when the stack was closed and the room was allowed to fill with smoke, permitting sampling
to be done over the course of several hours. The following analyses will focus on the "stack" burns, as those
measurements had little or no interferences from surfaces, where "room" burns are known to be compromised by the
loss of materials, such as $NH_3$, to the room walls at long sample times (Stockwell et al., 2014). Ash analyses were
performed only on the residues from the room burns and those values will be used for the N and C budget calculations,
with the assumption that stack and room burns left similar ash considering the combustion conditions were the same
for each type of fire. Table 1 lists the compounds and associated techniques used to measure them during the FireLab
2016 study, and describes the grouping of NVOCs measured by PTR-ToF into common categories, e.g. amines,
nitriles, etc.

**2.1 $N_r$ and NO measurements by Chemiluminescence**

Total reactive N ($N_r$) was measured by catalytic conversion to NO, followed by $O_3$-chemiluminescence using

an instrument described previously (Williams et al., 1998). $N_r$ and NO were sampled from inlets inserted adjacent to
the inlet-less open-path Fourier transform infrared spectrometer (OP-FTIR) instrument path during the stack burns
(Selimovic et al., 2018), and from a platform approximately 4 m off the floor in the middle of the room during the
room burns. The catalyst used for the $N_r$ channel, described in detail by Stockwell et al. (2018), consisted of a 11mm
I.D. quartz tube, packed with 36 platinum screens, heated to 750°C. This tube was wrapped with high temperature
heating tape and insulated inside a 7cm OD stainless steel tube that was fitted to a bulkhead placed through the wall





of the stack. The $N_r$ channel was diluted by a factor of 5:1 (±3%) using a flow of zero air added immediately
downstream of the Pt catalyst assembly. NO was sampled through a 6.3mm O.D. stainless steel inlet tube which was
placed through the bulkhead directly into the free air stream of the stack and connected to a 50mm Teflon filter holder
immediately outside the stack. The transfer lines for the $N_r$ and NO measurements consisted of 6.35mm O.D, 1mm
wall thickness PFA tubing of approximately 20 m in length. $N_r$ and NO data were acquired at 1 s frequency, but the
flow rate through each inlet was 1 SL min$^{-1}$, resulting in residence time in each inlet of 14 s. This time delay was
corrected in the data analysis. Any chemical effects of the inlet on the sampled air stream were negligible since the
analytes consisted of only NO and $NO_2$ and those are known to be transmitted by PFA Teflon tubing with essentially
no surface effects. However, there were possible effects of the inlets on the temporal features of the measurement
through diffusion or turbulent mixing. Those effects were examined through comparison of the temporal variations in
the NO signal with the NO measured by the OP-FTIR, and comparison of the $N_r$ signal under smoldering conditions
with the $NH_3$ measured by the OP-FTIR. Both of these comparisons showed that the NO and $N_r$ inlets had effective
time constants of 4 seconds, somewhat slower than the diffusive relaxation time assuming solely laminar flow.
The inlet streams were sampled into the NO instrument either directly (NO channel) or after passing through
a second catalyst of molybdenum oxide (MoOx) to convert remaining $NO_2$ to NO. The MoOx catalyst consisted of a
molybdenum tube at 350°C to which a small flow of $H_2$ (0.8%v/v) was added to control the re-dox state of the surface.
Both channels of the instrument were "de-tuned" to keep raw photon count rates below 4 MHz, by turning down the
$O_3$ flows and PMT voltages. Calibrations were performed with both a NO standard in $N_2$ (Scott-Marrin) and 10.1
ppmv standard of HCN in nitrogen (Gasco). The Pt catalyst was dismounted from the stack (or room) every few days
and checked for conversion efficiency by the addition of the HCN standard to the inlet. Conversion efficiencies were
found to be consistently high (>98%) throughout the entire sampling period (October 5 – November 12, 2016). There
were slight  background signals (a few tens of ppbv) for both NO and $N_r$ in both the stack and room air prior to and
after the burns, and those were subtracted from the fires signals prior to reporting the data. The overall uncertainties
in the NO and $N_r$ data were ±10% for each measurement.

**2.2 Other measurements**
Measurements of individual species during the 2016 FireLab study have been presented in several previous
publications. The OP-FTIR measurements were discussed by Selimovic et al., 2018, and the PTR-ToF measurements
were discussed by Koss et al., (2018). In addition, some of the calibration methods and GC separation and
identifications rely on additional analytical work presented by Sekimoto et al., (2017) and Gilman et al., (2015). We
measured the mass and elemental content of the initial fuel and the mass of unburned fuel for all the fires, and we
measured the mass and the elemental content of the ash during 21 room burns, which covered all the fuel types
discussed.

**2.3 PMF Analysis**
Trace gas measurements from multiple instruments involved in the FireLab study were combined and
analyzed using positive matrix factorization (PMF). PMF is a numerical method that was used in this case to partition




the compounds involved in a time varying mixture of chemicals into a few groups, or "factors", where a compound
can appear in more than one factor. A factor represents a consistent profile of compounds that is representative of one
of the sources contributing to the total signal. The sum of all the "factors" then ideally describes the total composition
of the measurements, which in this case is the emissions of $N_r$ compounds. By its nature, PMF assumes that the total
signal is a linear combination of individual sources that have a consistent composition, the relative contribution of
which is represented by the amount of each compound or category found in each factor (Paatero and Tapper, 1994;
Ulbrich et al., 2009). We hypothesize that species with dominant fractions in the same factor are related to each other
via the same formation processes. With knowledge of factor composition and the amount of each factor at any given
time the original emissions measurements can be reconstructed and this approach provides an alternate source of
profiles for fire emissions. PMF has also been used by a number of groups to explore how much various source profiles
contribute to complex ambient measurements (see for example Ulbrich et al., 2009) and was recently used to analyze
PTR-ToF-MS measurements from the FireLab (Sekimoto et al., 2018). Here, PMF was accomplished using the PMF
Evaluation Tool v. 2.08A (Ulbrich et al., 2009).

The application of PMF to this data set is different than the instances where it is applied to data from a single

instrument in which compound abundances are inherently scaled properly and error estimates are well defined and
self-consistent. For example, when applied to mass spectral data from a single instrument, errors can be expected to
scale as the square root of ion counts based on fundamental counting statistics (Sekimoto et al., 2018). In this work
we are including nitrogen measurements from several instruments, thus we chose to use mixing ratios as the unit of
comparison. The error estimates required by the PMF analysis were taken from the reported combined uncertainties:
the sum of the detection limit plus the estimated random error of the measured value. The variables that were used in
this PMF analysis and their units and corresponding errors are listed in Table 2. Where compound categories are
specified (e.g. nitriles), the values were the sum of the measured compounds in that category as listed in the footnotes
to Table 1. The data were further adjusted by subtracting the ambient air background before and after the fires, which
was a relatively minor adjustment for most compounds and categories. Any negative numbers that resulted were very
small compared to the fire emissions, and were set to zero.

Two approaches were taken when performing PMF analyses. The first approach included all individual N-

containing compounds together with $CO_2$, CO, and $N_r$ in the analysis batch (Batch 1), while the second excluded the
latter three species (Batch 2). $CO_2$ and CO were included because of their well-known roles as indicators of flaming
and smoldering combustion, respectively, an effect traditionally captured through the use of Modified Combustion
Efficiency (MCE) defined as

$MCE = \Delta CO_2/(\Delta CO_2 + \Delta CO)$                                    (Eq. 1)


where $\Delta CO_2$ and $\Delta CO$ are the $CO_2$ and CO levels above the ambient. When $CO_2$ and CO were included, the carbon
species were put into the PMF in units of ppmv, and all the nitrogen species in units of ppbv. This was done because
the nitrogen levels were on the order of a percent or less compared to the carbon species. The second analysis batch
(Batch 2) involved only the individually-measured nitrogen species and categories listed in Table 1 so that factor



loadings would be reflective of the nitrogen-only emissions. Batch 2 factors indicate how $N_r$ species are related to
each other via combustion chemistry.
We applied PMF to single fire data as well as extended time series that included all fires of a particular fuel
type, in-line with the approach laid out by (Sekimoto et al., 2018). By consolidating fuels from a particular vegetation
type, the fire to fire variability largely driven by differences in the fuel (e.g. moisture content, structure, quantity) is
constrained and the most representative fire conditions are captured. Two fuel groups were analyzed in this way: the
western U.S. coniferous ecosystem fuels which included ponderosa pine, lodgepole pine, Douglas fir, Engelmann
spruce, and sub-alpine fir and the chaparral ecosystem which was represented by chamise and manzanita. The
consolidated time series for the coniferous ecosystems included realistic mixtures, canopy only, and litter only, while
duff and rotten logs were analyzed separately, and not included in the timeseries.

**3 Results and Discussion**
Example timeseries of NO, $N_r$, $\Delta CO$, $\Delta CO_2$ (CO and $CO_2$ corrected for their backgrounds) are shown in
Figure 2, for a fire burning a sample of ponderosa pine realistic mix (Fire 004).  MCE was also plotted in addition to
the chemical species. The timeseries for Fire 004 shows a short initial smoldering/distillation phase (MCE 0.7 to 0.8)
as heat pyrolyzes the fresh fuel and releases VOCs from exisiting pools in the fuel followed after ignition by a
relatively efficient mix of flaming and smoldering combustion (MCE 0.95 to 0.98) and then finally a subsequent
period of essentially pure smoldering (MCE ~0.80). The $N_r$ and NO timelines had many features in common because
NO is often the most abundant $N_r$ compound (see below). As a result, it is useful to compare the quantities $N_r$–NO
and $(N_r-NO)/N_r$ to the other measures of chemical species or combustion efficiency. As expected, $(N_r-NO)/N_r$, in
Figure 2(c) is anti-correlated with MCE since $N_r$ is primarily NO at high MCE. In addition to the anti-correlation, this
non-NO fraction, like its approximate carbon analog $CO/CO_2$, has a wider dynamic range than MCE and will often
suffer less from background variability than carbon-based indices (Yokelson et al., 2013a).
The concentration profiles of the background-corrected measurements of $N_r$, $CO_2$, CO, and all the carbon-
containing species measured by the FTIR (Selimovic et al., 2018) during the stack burns were integrated over the
entire time of the burn to obtain total carbon, termed TC here, and total $N_r$. The additional carbon species included
methane and a number of other gas phase VOCs as well as organic- and black-carbon aerosol. Altogether these carbon
species should account for ≥98% of emitted carbon (McMeeking et al., 2009). Total $N_r$ is plotted in Figure 3, versus
TC (Figure 3a) and versus nitrogen burned, which is calculated from the %N in the fuel times the mass of fuel
consumed (Figure 3b). The points in Figure 3 are colored by the fuel N/C mole % obtained from the elemental analysis
of each fuel. The relationship between $N_r$ and TC in panel 3a clusters around the 0.37% line and those points are from
fuels most characteristic of the North American biomes impacted by wildfire. There are clear outliers in the correlation
of $N_r$ and TC; for example, yak dung and two samples of duff were high due either to the fact that they have high fuel
N/C ratios, or they burned with minimal flaming (whole fire MCEs 0.86-0.89), hence experienced less de-nitrification.
The fuels that were low in $N_r$/TC in panel 3a, ponderosa pine rotten log, subalpine fir and excelsior, had low fuel N/C,
so when plotted versus nitrogen burned in panel 3b, they cluster with the main group of characteristic fuels, i.e. they
are no longer 'outliers' in the distribution.





The points in Figure 3a are all lower than the corresponding fuel N/C mole ratio, due to the denitrification
chemistry, shown in Figure 1, and verified in lab studies described by Kuhlbusch et al. (1991), and the production of
$N_2O$ which is also not measured by the $N_r$ technique. The sum of $N_2$ and $N_2O$ produced in the fires can be estimated
from the difference between the fuel N/C and the $N_r$/Total C emitted and the data on C and N content remaining in
the ash. The mass balance equations used for these calculations are detailed in the Supplemental Materials.
The distribution of the N lost to $N_2$ and $N_2O$ is shown in Figure 4. Chemical analyses were not done for all
fuels during the stack burns, and the analysis above assumes that the ash residues and ash/burned fuel ratios from the
stack burns were well represented by those for the same fuels used in the room burns, for which mass yields and
chemical analyses were done. Data are missing for fuels that did not have a corresponding ash analysis. The median
fraction of N lost to $N_2$ and $N_2O$ for ash-corrected fires was 0.70, and the mean (±standard deviation) was 0.68 (±0.14).
This fuel-based estimate is uncertain by approximately 25% because of the above assumptions concerning the
applicability of the ash analyses from the room burns and because fuel moisture corrections were assumed to apply to
all of the materials burned, foliage vs. woody biomass (see SI for details). The emission of $N_2O$ relative to $N_2$ is
approximately 10% or less for a wide range of fuels (Andreae, 2019). Assuming the N remainder in our work is at
least 90% $N_2$ gives values that are somewhat higher than the $N_2$ values reported by Kuhlbusch et al., (1991) where $N_2$
accounted for 36% of fuel N burned in flaming stage fires. A closer inspection of Kuhlbusch et al., (1991) showed a
range of $N_2$ yields of 40-54% at highest MCEs of 0.94-0.97. Possible reasons for these differences are that the
Kuhlbusch et al., (1991) fires were limited to grasses, hay, and pine needles, and the fires were confined to a closed
container and so may not have experienced the convection and turbulence of typical biomass fires. In addition, the
fires analyzed in our work were somewhat weighted towards the full canopy and higher temperature burning fuels,
since ash analyses were not done for peat, dung and many of the "litter" samples, all of which tend to burn less
efficiently. Goode et al., (1999) estimated an $N_2$ emission of 45±5% for MCE values of 0.95 in grass and surface fuels.
The range of values determined in our work overlap with these literature values, but are on average higher. It should
be noted that such loss of reactive nitrogen can have implications for ecosystem N budgets, as discussed by Kuhlbusch
et al., (1991).
The composition of the N that does not get converted to $N_2$ or $N_2O$ is of intense importance in determining
atmospheric impacts of fires. Emission factors for all the individual $N_r$ compounds identified in our work have been
compiled and reported in previous publications (Koss et al., 2018; Selimovic et al., 2018), so this paper will focus on
the $N_r$ budget. The balance of $N_r$ budget for Fire 047, sub-alpine fir realistic mix, is shown in Figure 5, in which the
timelines of $N_r$, NO, $N_r$-NO, sumN, and NVOC are plotted along with MCE and $(N_r-NO)/N_r$. The quantity sumN is
the sum of all other non-NO compounds, and NVOC is the subset of sumN that are organic nitrogen compounds
measured by the PTR-ToF, as listed in Table 1. This fire had a mixture of flaming and smoldering combustion
throughout the fire as indicated by MCE and nitrogen profiles (panel (d)). The comparison of $N_r$-NO with the sumN
in panel (b) shows that much of the N is accounted for. The major contributors to sumN for this fire were HNCO,
HCN, HONO, $NO_2$, and $NH_3$, while NVOC was a very small contributor to sumN (panel (b)). Note that while $HNO_3$
is measurable by FTIR with good sensitivity, no $HNO_3$ signals were observed above detection limit, which was a few
ppbv. Panel (c) shows the residual left after NO and sumN are subtracted from $N_r$, corresponding to an integrated



amount of 15.6 ±8% of $N_r$. This residual is reasonable considering typical published particle $N_r$ measurements (Akagi
et al., 2012; Akagi et al., 2011; Liu et al., 2017; May et al., 2014), and consistent with there being some particle $N_r$
from flaming, which are most likely organic nitrates or nitro-organics, and particle ammonium from smoldering with
potassium or ammonium nitrate potentially accounting for substantial $N_r$.

In contrast to the above, the nitrogen emissions from Fire 050, yak dung, are shown in Figure 6. This fuel

produced mostly smoldering emissions as exemplified by the low NO levels relative to $N_r$ (panel a), and the low MCEs
observed (panel d). The sum of $N_r$ species was somewhat correlated with the quantity $N_r$-NO, but was substantially
lower, and the residual $N_r$ unaccounted for by the gas-phase measurements was 33.9 ±16% of $N_r$ (panel c). The
majority of sumN was represented by HCN and $NH_3$, with acetonitrile ($CH_3CN$) higher than any of the other
inorganics, HNCO, $NO_2$ or HONO. The NVOCs were also a larger fraction of $N_r$-NO than in the case of Fire 047
shown above, a feature that implies that more semi-volatile organic compounds, SVOC, survive these types of fires
and could make a proportionally higher contribution to the $N_r$ budget in this fire relative to Fire 047. FireLab results
of particle organic carbon measurements (Jen et al., 2019) and field measurements in environments with a lot of dung
burning (Jayarathne et al., 2018; Stockwell et al., 2016a) are consistent with a higher EF for particle organic carbon
and by extension particle NVOC compounds. The quantity ($N_r$-NO)/$N_r$ was relatively high and had less dynamic range
than for fires with more flaming combustion like Fire 047.

The fire-integrated measurements of inorganic and NVOC species are listed in the Supplemental section as

ratios to $N_r$ for each stack fire (Table S1). The summary of all the fire integrated $X_i/N_r$ fractions (where $X_i$ is the $N_r$
species or quantity) is given in Table 3 for all the fires for which we have a complete set of measurements (43 fires).
In general, NO was the major species followed by $NH_3$, and the other inorganic $N_r$ species, $NO_2$, HNCO, HONO, and
HCN had individual contributions of 4.3 to 9.4 %. NVOC species were less than 5% of $N_r$ on average. The
unaccounted-for $N_r$, defined as ($N_r$-NO-sumN)/$N_r$ had a median value of 14.3% and a mean (±std. dev.) of 15 (±10)%.
Overall, 85% of $N_r$ was accounted for by the gas phase measurements. The distribution of whole fire $N_r$ residuals is
plotted as a histogram in Figure 7. We expect the residual $N_r$ was composed of either semi- or low-volatility
compounds, or particle-bound $N_r$ compounds, which we know are converted efficiently by the $N_r$ catalyst (Stockwell
et al., 2018) but not detected by the instruments included in this analysis. Along these lines, there is some indication
that the residual has a systematic variation with whole fire MCE, with higher residuals (up to 30%) observed at lower
MCEs and higher ($N_r$ – NO)/$N_r$ (see Figure S1 a&b), which would be consistent with higher EF for SVOC at low
MCE (Jen et al., 2019) and particle $N_r$ having a higher contribution from $NO_3^-$ (May et al., 2014), and perhaps particle
ammonium or reduced-$N_r$ compounds. In general, there is more particulate organic material emitted from fires at low
MCE (Jen et al., 2019), so we would expect more particle N at low MCE to go along with that.

**3.1 Systematic dependences of $N_r$ composition on combustion processes.**

The features noted in fires shown above, as well as the anti-correlation of MCE and ($N_r$-NO)/$N_r$ lead to the

question of whether there are systematic dependences in $N_r$-compound composition on fire stage that can be used to
formally classify and/or potentially predict the relative emissions of $N_r$ compounds. MCE has been used as a rough
indicator of the relative amounts of flaming and smoldering combustion in a fire, with high MCE (~99%) being "pure"



flaming, low MCE (~80%) being "pure smoldering," and an MCE of ~0.9 being roughly equal amounts of both (Sect.
2.1.1 in Akagi et al., 2011). It should be understood that "smoldering" in this framework is a lumped term that includes
all non-flame processes such as pyrolysis, glowing, and distillation, while flames cannot exist without these processes
producing gaseous fuel to support them (Yokelson et al., 1996). In addition, "pure flaming" is essentially the efficient
oxidation of smoldering products before they enter the atmosphere. However, for MCE to predict flaming and
smoldering $N_r$ species well, the variable fuel N must be considered. For instance, $NO_x$ is clearly produced by flaming
based on its temporal profile, but fire-integrated $EF_{NOx}$ may not correlate with MCE due to variable fuel N. In these
cases, $EF_{NOx}$/fuel N or $\Delta NH_3/\Delta NOx$ may still correlate (or anti-correlate) well with MCE (e.g. Fig. 4 in Burling et al.,
2010 or Yokelson et al., 1996). Finally, the flame chemistry involving $NH_3$, HNCO, and HCN both produces and
destroys NO in a fashion that does not conserve $N_r$. This chemistry is explored in Figure 8 in which $NOx$, $NH_3$, HNCO,
HCN, HONO, and $CH_3CN$ ratios to $N_r$ are plotted vs real-time MCE for Fire 047 as a typical example for fires that
have a substantial range of MCEs (e.g. from 0.8 to above 0.98). The relationship between $NH_3/N_r$ and MCE confirms
that $NH_3$ is primarily a smoldering emission and $NO_x/N_r$ increases with increasing MCE in a non-linear fashion that
confirms it is primarily a flaming compound. Such a non-linear dependence has also been seen for other flaming-
related quantities such as Elemental Carbon/TC or $EF_{HCl}$ (Christian et al 2003; Stockwell et al., 2014). Most
importantly, the variations of $HNCO/N_r$, $HCN/N_r$, $HONO/N_r$, and $CH_3CN/N_r$ versus MCE don't arise dominantly
from either regime as these are species that are likely produced by multiple pathways (e.g. "incomplete flaming",
pyrolysis, possibly glowing). By "incomplete" flame chemistry we mean the production of incompletely oxidized
products in flames such as the complex system of reactions shown in Fig. 1. These reactions involving HNCO, HCN
and $NH_3$ both produce and destroy NO, while HONO is produced from reactions of NO and $NO_2$ that are faster at
slightly lower temperatures, for example the three-body association reaction of NO with OH radical (Manion et al.,
2015). Variable turbulence in the turbulent diffusion flames that are characteristic of open BB likely contributes to
varying temperatures, and therefore, varying amounts of incomplete oxidation of the fuel N (Shaddix et al., 1994).

The complexity of the dependence of $N_r$ speciation on combustion chemistry suggests that MCE is an
insufficient model to use for applying lab results to real-world fire emissions (Stockwell et al., 2016a; Yokelson et al.,
2013b). Accordingly, we employed the positive matrix factorization (PMF) method (see Methodology section) that
has been used by a number of groups to probe the sources contributing to complex mixtures (see for example Ulbrich
et al., 2009 Sekimoto et al., 2018). Our PMF results showed several general features, irrespective of the inclusion or
exclusion of $CO_2$, CO and $N_r$. The emissions were best fit by three factors (with approximate descriptive names
justified below and prime species): (1) a combustion (flaming) factor (abbreviated Comb-N), (2) a high temperature
pyrolysis factor (HT-N), and (3) a low temperature pyrolysis factor, (LT-N). We use these terms in part to harmonize
our discussion with the VOC results discussed by Sekimoto 2018. An example timeseries for the PMF analysis of a
coniferous fuel with just the $N_r$ species included is shown in Figure 9 for a realistic mix of lodgepole pine (Fire 063),
and Figure S2 shows the consolidated time series of all coniferous fuels fit using just the $N_r$ species. The three factors
successfully describe the majority of the $N_r$-emissions where the difference between the measured and calculated mass
is on average 5.1% for coniferous fuels and 4.6% for chaparrals as indicated in Table 4.





The 'loadings' of the three different factors, i.e. the contribution of compounds to each factor, for coniferous
fuels are shown in Figure 10(a), and the distribution of a given compound or compound class amongst the three factors
is shown in Figure 10(b) as normalized fraction. Normalized fraction is equal to the PMF-determined contribution of
a compound to a factor, divided by the sum of the contribution of the compound to all three factors. The Comb-N
factor contained NO, $NO_2$, and HONO, the HT-N factor had mostly HCN, HNCO, nitriles, with contributions from
$NO_2$ and nitro compounds, and the LT-N factor contained $NH_3$, amines, amides and heterocyclics. Within the Comb-
N factor there is some evidence that the relative amounts of HONO and NOx depend on fuel moisture. For example,
the ratio HONO/NOx for whole fires shows some correlation with needle moisture in coniferous fires that were canopy
fuels (Foliage and small woody biomass), as shown in Figure S3. This may be due to flame process the interconvert
NOx and HONO in the presence of water vapor of OH (see Figure 1).
Literature values from studies where flame temperature was measured are typically in the range of 1100 –
1200 °C (Taylor et al., 2004; Wotton et al., 2012), so we would assume that would constitute the upper range of our
Comb-N factor. The radical chemistry involving HCN, HNCO and $NH_3$ starts to shut down below about 800-900°,
according to the modeling of Glarborg et al., (2018), so we set 800°C as a lower limit for the Comb-N factor. The HT-
N factor species are known to be produced by the intense pyrolysis of fuel $N_r$ compounds (Hansson et al., 2004; Liu
et al., 2018; Ren et al., 2010), which for these compounds becomes important at temperatures around 500-600°C.
Accordingly, we estimate the temperature range for the HT-N factor at 500 – 800°C. The remaining LT-N factor
results from mild pyrolysis and pertains to fire conditions of roughly 500°C and below, and was dominated by $NH_3$,
amines, amides and some of the more complex organics (Koss et al., 2018). The names and temperature ranges are
approximate and likely include processes that occur inside flames as part of the flame proper, as turbulent diffusive
flames are highly variable in space and time.
It is useful to explore the correlation of N-PMF factors with other fire indicators to determine relationships
for parameterizing $N_r$ emissions together with carbon and VOC emissions. The Comb-N factor for coniferous fuels,
which consisted of NOx and HONO, would be expected to correlate with $CO_2$ but not as well with MCE since the
latter includes an indicator of incomplete combustion. The timeseries of Comb-N along with $CO_2$ and with MCE for
Fire 037 (ponderosa pine), are plotted in Figure 11. As expected they show an excellent correlation of Comb-N with
$CO_2$ ($R^2$=0.942) since all the species are flaming compounds, but non-linear correlation of Comb-N with MCE
($R^2$=0.363) since the latter factors in a smoldering compound (CO), similar to the NOx/$N_r$ vs. MCE plot for Fire 047
in Figure 8. The excellent correlation of Comb-N with $CO_2$ is a broadly applicable result, the $R^2$ parameters for all the
fires shown in Figure S2 had an average of 0.898, and ranged from 0.806 to 0.966. As a consequence, we can conclude
that $CO_2$ would be the best tracer for Comb-N in many western U.S. ecosystems where conifers predominate, provided
ambient $CO_2$ backgrounds can be properly accounted for as described by Yokelson et al., (2013a).
Our Comb-N factor did not correspond to the high temperature VOC factor (HT-VOC) found by Sekimoto
et al., (2018), because combustion produces NOx, and HONO, but almost none of the compounds classified as VOCs
survive these conditions. However, the HT-N and HT-VOC factors were well correlated for many fires. An example
of this is shown in Figure 12 for Fire 037, a sample that was broadly representative of ponderosa pine (i.e. canopy and
litter). This result can be rationalized by the fact that while HT-VOC factors have large contributions from many more



compounds that the N compounds measured here, they also have large contributions (>85%) from HCN, HNCO, and
HONO, (in other words >85% of HCN, HNCO and HONO are found in the HT-VOC factor). Since the HT-N factors
are also heavily weighted by HCN and HNCO, it is reassuring that both of these PMF analyses have independently
identified these species as important contributors to the HT fire regime. The $R^2$ correlation coefficients of HT factors
for the coniferous shown in Figure S2 averaged 0.866 and ranged from 0.419 to 0.959. As a consequence of this
correlation, we can conclude that HCN is the best marker for the HT-N and HT-VOC factors in most western U.S.
wildfires, since HCN is essentially inert on the timescales of fire plumes.

The correlations of LT-N and LT-VOC factors were not particularly high for most of the coniferous fires

shown in Figure S2. The average $R^2$ was 0.427 with a range of between 0.072 and 0.827. The reasons for this lack of
correlation are not clear, as $NH_3$, amines and amides appear predominantly in both LT factors, and the absolute
concentrations of $NH_3$ are usually quite high in these fires relative to VOCs (Sekimoto et al., 2018). However, the LT-
VOC factor includes many more compounds with a variety of functional groups not found in the LT-N factor, so it
appears that the VOC and N compounds have sufficiently different pyrolysis chemistry that the LT factors do not
show much correlation. We conclude that $NH_3$ (and particle $NH_4^+$) will be the best marker for the LT-N factor in
western U.S. coniferous wildfires, but the LT-VOC chemistry might not be captured reliably by this marker.

The emissions from burning chaparral fuels (manzanita and chamise) collected at two sites in California were

also analyzed as a group and yielded three separate factors in a fashion similar to the coniferous fuels (see Figure S4
for the PMF timeline). The chaparral factors had slightly different composition (Figure S5), the combustion factor
was mostly NO, with small amounts of HNCO, HONO and $NH_3$, the high temperature factor was dominated by $NO_2$
and included HONO, HCN, and HNCO, and the low temperature factor was mostly $NH_3$ with a slight amount of NO
contributing. The NVOC species were found in both the medium and low temperature factors.

There was less similarity between the Comb-N factor and $CO_2$ emissions for chaparral fuels compared to

those found for coniferous fuels, with an average correlation coefficient ($R^2$) of 0.689, with a range from 0.244 and
0.950. As a result, there may not be a simple conserved tracer for the combustion factor of these fuel types, however
total odd nitrogen (NOy) which is NOx and all the compounds that are produced from NOx in the troposphere, may
be useful as it is a reasonably conserved tracer in the absence of wet or dry deposition of particles. Correlation
coefficients between the HT-N and HT-VOC factors were on average $R^2 = 0.551$, with a range 0.047-0.911. The
correlations between LT-N and LT-VOC factors were in the same range for chaparral fuels as for coniferous, average
$R^2 = 0.447$, range 0.028-0.827.

There were some fuels that do not sustain flaming combustion well, specifically duff, Yak dung and

Indonesian peat. These fires exhibited little or no NO emission commensurate with minimal flaming combustion.
Instead the emissions were mostly the pyrolysis products $NH_3$, (0.22 – 0.53 $N_r$ fraction), and HCN (up to 0.32 $N_r$
fraction for peat). It was also apparent that these fires also had unaccounted for $N_r$, close to, or just over 0.30 (Table
S1). The distribution of $N_r$ compounds in the one peat fire that we measured (Fire 055) is in line with those reported
for fires measured in situ which showed relatively high EFs for HCN and $NH_3$ (Stockwell et al., 2016b; Stockwell et
al., 2015).



The application of our $N_r$ emissions results to real-world fires will depend somewhat on the nature of the

information available on a particular fire, or fire complex. As a good starting point, or in the absence of detailed N

and C analyses of fuels, a $N_r$/C ratio of 0.37% appears to capture most of the fires studied in this work. The Nr can be

apportioned according to the results summarized in Table 3. Adjustments to those fractions can be made either by

scaling slightly by average MCE, with the knowledge that intermediate species (HT-N pyrolysis species) such as HCN

and HNCO do not scale in the simple manner that $NH_3$ and $NO_x$ do. If measurements of marker compounds are

available then $CO_2$, HCN, and the sum $NH_3 + NH_4^+$ can be used for the combustion, high-temperature pyrolysis, and

low-temperature pyrolysis factors respectively.

**4 Conclusions**

Seventy-five stack fire experiments were conducted during the FIREX FireLab experiments in Fall, 2016. A

range of fuels characteristic of the western U.S. was burned under conditions and in mixtures meant to represent

authentic wildfire conditions, as closely as is possible in the laboratory. Total reactive nitrogen ($N_r$ = all N-containing

compounds except $N_2$ and $N_2O$) was measured along with a suite of N-containing compounds in order to obtain a

budget for $N_r$-emissions and to examine relationships between fuels, combustion conditions, and emissions chemistry.

Natural convection wildfires do not burn hot enough to produce $NO_x$ from $N_2$ and $O_2$, so all $N_r$ emissions

come from the fuel N. Almost all of the fires representative of North American ecosystems had emissions that clustered

around a $N_r$/C ratio of 0.37%, which can serve as a starting point for scaling emissions from these ecosystems

Comparison total $N_r$ and total carbon emissions with the N/C ratios of both the original fuel and remaining ash allowed

468        us to estimate that an average of 68% (±14%) of the fuel nitrogen ends up as $N_2$ and $N_2O$. This loss of nitrogen can

be used to estimate how much fuel nitrogen ends up as $N_r$, which is a crucial aspect of fire plume chemistry since the

photo chemistry many fire plumes is NOx-limited, and $NH_3$ is an important contributor to particle chemistry. Of the

remaining N emitted as $N_r$, approximately 85% (±10%) was accounted for by individually measured gas-phase species,

while the rest was most likely particle-bound $NH_4^+$ and $NO_3^-$, with a smaller contribution from low-volatility species

or other species such as cyanogen (Lobert and Warnatz, 1993), that were not quantified by the instruments for

individual measurements we used in this study.

The individual $N_r$ species composition normalized to Total $N_r$, to account for fuel N variability, correlated

with flaming versus vs. smoldering combustion as indicated by modified combustion efficiency (MCE) for some

species (e.g. $NH_3$, $NO_x$). Other species, such as HCN and HNCO, peaked at intermediate MCE values. However,

positive matrix factorization (PMF) showed that all the measured $N_r$ emissions from the main two categories of fuels,

conifers and chaparral, grouped into three mixtures (factors), roughly attributed to temperature: combustion (NOx,

HONO), high temperature (HNCO, HCN, nitriles), and low temperature ($NH_3$, amines, amides). Chemical kinetic and

pyrolysis considerations set the temperature ranges for these regimes at approximately 800-1200°C, 500-800°C and

<500°C respectively.

This paper connects mechanistic aspects of N combustion chemistry to the budget of $N_r$ emissions from

biomass burning. The emission composition measurements detailed here give useful information concerning what the

initial conditions will be in actual fire plumes. These results suggest that for coniferous fuels characteristic of the



western U.S. $CO_2$ is the best marker for flaming combustion, HCN is the best marker for high temperature pyrolysis processes, and $NH_3/NH_4^+$ is the best marker for low temperature pyrolysis processes. The HT-N and HT-VOC pyrolysis factors showed high degree of correlation especially for coniferous fuels, which can simplify how these different classes of emissions can be estimated. Results of future field intensives can be combined with this emissions information to refine these recommendations on how to put $N_r$-chemistry into the modeling frameworks needed to predict fire plume chemistry and impacts.

**Data availability**

The FIREX Firelab 2016 data are available at: https://esrl.noaa.gov/csd/groups/csd7/measurements/2016firex/FireLab/DataDownload/. The descriptions of the measurements can be found here: https://esrl.noaa.gov/csd/groups/csd7/measurements/2016firex/FireLab/dataidtable.html. The complete ash analyses are available on request.

**Author Contributions**

JMR, RY, CW and JdG designed the research. The measurements were conducted by JMR, CS, CW, RJY, JdG, YL, VS, ARK, KS, MMC, BY, KJZ, SSB, CS, and SHD. All authors contributed to the discussion and interpretation of the results and writing the paper.

**Competing interests**

Joost de Gouw worked as a consultant for Aerodyne Research during part of the preparation phase of this paper.

**Disclaimer**

Any mention of brand names or manufacturers is for information purposes only and does not constitute an endorsement.

**Acknowledgements**

A. Koss acknowledges funding from the NSF Graduate Fellowship Program. K. Sekimoto acknowledges funding from the Postdoctoral Fellowships for Research Abroad from Japan Society for the Promotion of Science (JSPS) and a Grant-in-Aid for Young Scientists (B) (15K16117) from the Ministry of Education, Culture, Sports, Science and Technology of Japan. R. Yokelson and V. Selimovic were supported by NOAA-CPO grant NA16OAR4310100. J. de Gouw was supported by the NSF AGS grant 1748266 under a subcontract to the University of Montana during the analysis phase of this work. We thank the USFS Missoula Fire Sciences Laboratory for their help in conducting these experiments, especially Shawn Urbanski and Thomas Dzomba. This work was also supported by NOAA's Climate Research and Health of the Atmosphere Initiative.

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






**Table 1. Nitrogen compounds observed in the FIREX FireLab 2016 Study.**


| Compound/Class | Importance | Measurement Method | References |
|---|---|---|---|
| Total Reactive N | Total available for atmospheric reactions | Catalytic Conversion NO/$O_3$ chemiluminescence | Stockwell et al., 2018 |
| Nitric Oxide | Major "flaming stage" product, oxidant production | NO/$O_3$ chemiluminescence OP-FTIR | Williams et al., 1998 Selimovic et al., 2018 |
| Nitrogen Dioxide | Atmospheric oxidant production | OP-FTIR, ACES | Stockwell et al., 2014, Min et al., 2016, Zarzana, et al., 2018 |
| Nitrous Acid | $HO_x$ radical source | OP-FTIR, ACES | Stockwell et al., 2014, Min et al., 2016, Zarzana et al., 2018 |
| Nitric Acid | Particle precursor | OP-FTIR | Yokelson et al 2009, McMeeking et al 2009 |
| Hydrogen Cyanide | Flame chemistry, Atmospheric tracer, Toxicity | OP-FTIR, PTR-ToF-MS | Selimovic, et al., 2018 Koss, et al., 2018 |
| Isocyanic Acid | Flame chemistry, Toxicity, Health effects | PTR-ToF-MS | Koss, et al., 2018 |
| Ammonia | Major "smoldering stage" product, Main atmospheric base, Particle formation | OP-FTIR | Selimovic, et al., 2018 |
| NVOCs: Amides[1] Amines[2] Heterocyclics[3] Nitriles[4] Nitro compds[5] | Brown carbon, Toxicity, Tracers | PTR-ToF-MS, GC/MS, I⁻ CIMS | Koss, et al., 2018 Gilman et al., 2015 Lerner et al., 2017 Lee et al., 2014 |

1). Ethylamine, methanimine, propeneamine, sulfinylmethanamine, trimethylamine, buteneamines
2). Formamide, acetamide, methylmaleimide
3). $C_2$-pyrroles, dihydropyridine, ethynylpyrrole, methylpyridine, methylpyrrole, pyridinealdehyde, 4-pyrindinol,
vinylpyridine
4). Acetonitrile, acrylonitrile, benzonitrile, butanenitrile, butynenitrile, benzoacetonitrile, $C_7$acrylonitrile, $C_8$-nitriles,
heptylnitrile, furancarbonitrile, methylbenzoacetonitrile, pentylnitriles, propanenitrile, propynenitrile, butenenitrile,
methylisocyanate.
5). Butenenitrates, nitrobenzene, nitroethane, nitroethene, nitrofuran, nitromethane, nitropropanes, nitrotoluene.





**Table 2. Compounds and compound classes used in the PMF analyses and their**
**corresponding errors.**

| Compound or Class | unit | Batch 1 | Batch 2 | Estimated error |
|---|---|---|---|---|
| $CO_2$ | ppmv | X | | 20% + 2 ppmv |
| CO | ppmv | X | | 20% + 0.002 ppmv |
| $N_r$ | ppbv | X | | 10% + 1 ppbv |
| $NH_3$ | ppbv | X | X | 5% + 2 ppbv |
| NO | ppbv | X | X | 10% + 1 ppbv |
| $NO_2$ | ppbv | X | X | 10% + 0.2 ppbv |
| HONO | ppbv | X | X | 20% + 1 ppbv |
| HCN | ppbv | X | X | 15% + 0.2 ppbv |
| HNCO | ppbv | X | X | 15% + 0.2 ppbv |
| Nitriles | ppbv | X | X | 20% + 0.2 ppbv |
| Amines | ppbv | X | X | 20% + 0.2 ppbv |
| Amides | ppbv | X | X | 20% + 0.2 ppbv |
| Nitro-compounds | ppbv | X | X | 20% + 0.2 ppbv |
| Heterocyclics | ppbv | X | X | 20% + 0.2 ppbv |

**Table 3. Summary of $X_i/N_r$ Measurements for all Stack Burns[1]**

| Quantity | Average ±(std dev) % |
|---|---|
| $NO/N_r$ | 34.5 (16.6) |
| $NO_2/N_r$ | 9.4 (6.2) |
| $HNCO/N_r$ | 6.0 (2.9) |
| $HONO/N_r$ | 4.5 (2.2) |
| $HCN/N_r$ | 4.3 (2.3) |
| $NH_3/N_r$ | 19.3 (6.7) |
| $NVOC/N_r$ | 4.3 (2.8) |
| $(N_r-sumN)/N_r$ | 15.2 (9.8) |

1). Not every measurement was available for every fire, consequently the values do not add up to
exactly 100%.



**Table 4. Residuals of the PMF analyses by fuel, as percent of total signal**

| Fuel | Total Number | Component | Fire Number | Residual (%), avg (stdev) |
|---|---|---|---|---|
| Ponderosa Pine | 9 | Realistic (mix) | Fire 37,59,72 | 3.8 (± 1.4) |
|  |  | Canopy (pure) | Fire 19[a],39 |  |
|  |  | Litter (pure) | Fire 38 |  |
| Lodgepole Pine | 5 | Realistic | Fire 07[a],58,63 | 5.1 (±3.1) |
|  |  | Canopy | Fire 40 |  |
|  |  | Litter | Fire 41 |  |
| Douglas Fir | 4 | Realistic | Fire 14[a],57 | 6.8 (±3.1) |
|  |  | Canopy | Fire 18 |  |
|  |  | Litter | Fire 43[a] |  |
| SubAlpine Fir | 5 | Realistic | Fire 47,67 | 6.6 (±2.3) |
|  |  | Canopy | Fire 15,23 |  |
|  |  | Litter | Fire 51[a] |  |
| Engelmann Spruce | 2 | Realistic | Fire 08[a] | 3.1 (±1.9) |
|  |  | Canopy | Fire 25 |  |
| Chamise (San Dimas, CA) | 2 | Canopy | Fire 24,29 | 4.4 (±2.7) |
| Chamise (North Mountain, CA.) | 2 | Canopy | Fire 27,32 | 4.2 (±1.0) |
| Manzanita San Dimas, CA) | 2 | Canopy | Fire 30,33 | 4.8 (±2.1) |
| Manzanita (North Mountain, CA.)) | 2 | Canopy | Fire 28 | 5.1 |

a-Excluded from Batch 2




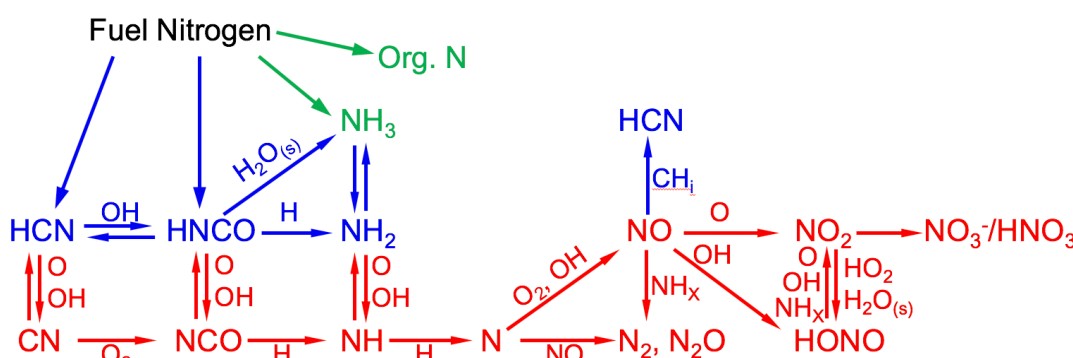

**Figure 1. Schematic of the combustion chemistry of the small molecules that are emitted**
**from BB and represent sources and sinks of reactive nitrogen ($N_r$), adapted from (Glarborg**
**et al., 2018; Lobert and Warnatz, 1993; Lucassen et al., 2012). $H_2O_{(s)}$ denotes the combination**
**of $H_2O$ and a surface to facilitate the reaction. Red color indicates the highest temperature**
**(combustion) processes, blue indicates intermediate temperature processes and green**
**indicates the lowest temperature processes.**


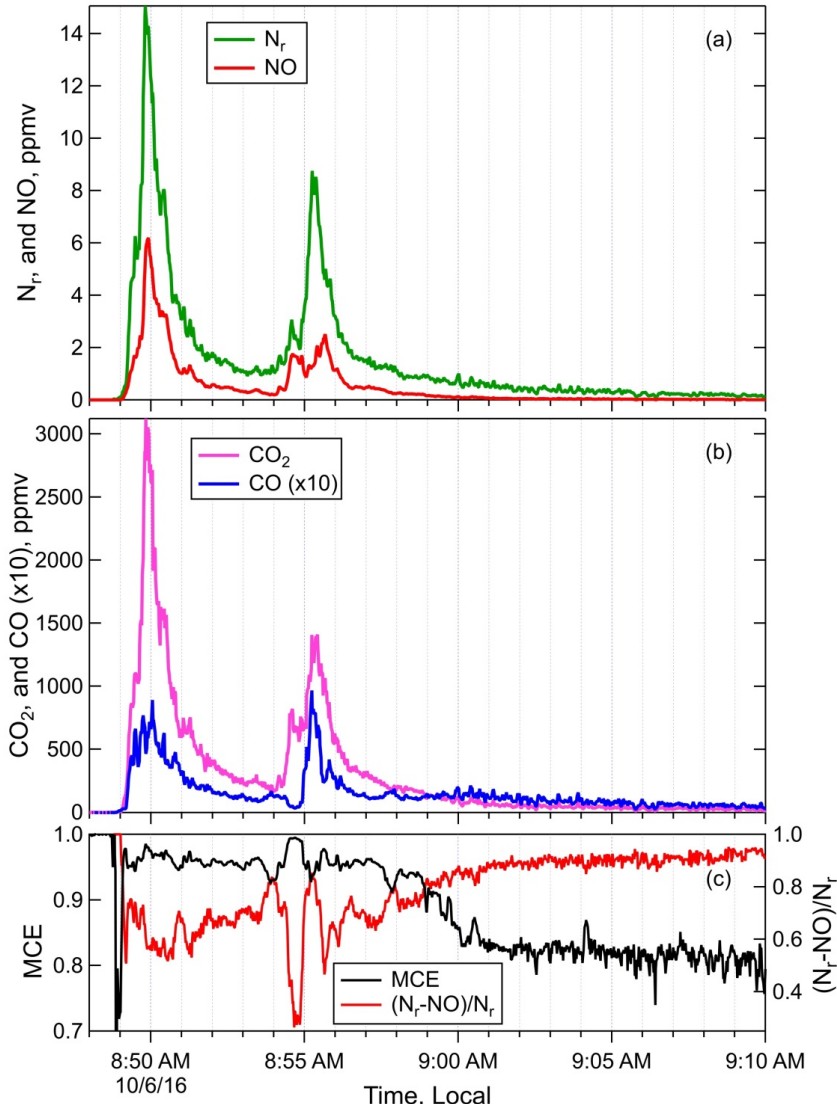



**Figure 2. Timelines of the $N_r$, NO (panel a), $\Delta CO_2$, $\Delta CO$ (panel b), MCE and $(N_r–NO)/N_r$ (panel c) measured during Fire 004, a ponderosa pine realistic mix sample. Note that $\Delta CO$ is plotted at x10 the measured abundance for clarity.**





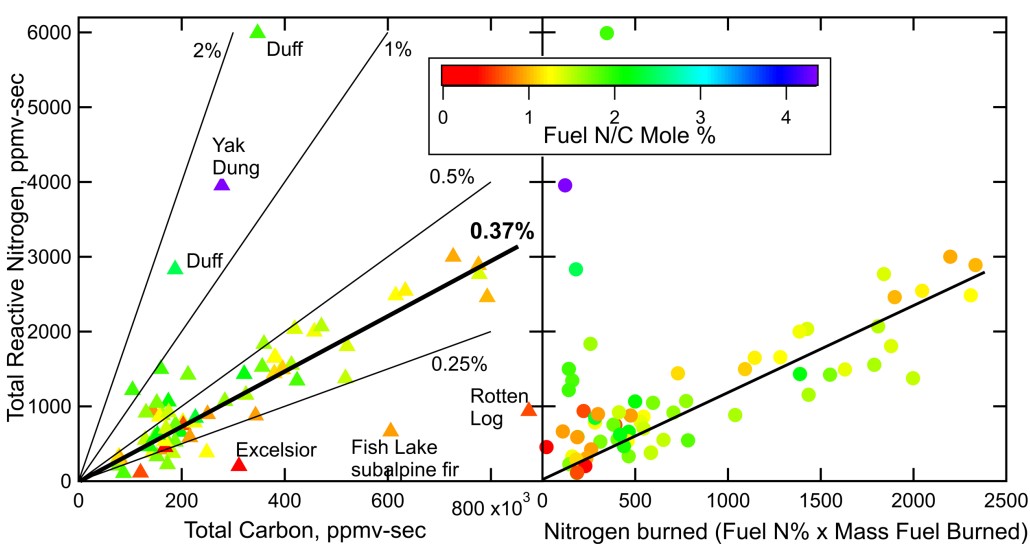

**Figure 3. Integrated $N_r$ versus integrated Total Carbon (panel a), and versus nitrogen**
**burned based on fuel nitrogen content and mass of fuel burned (panel b). The points are**
**colored by fuel nitrogen to carbon ratio.**




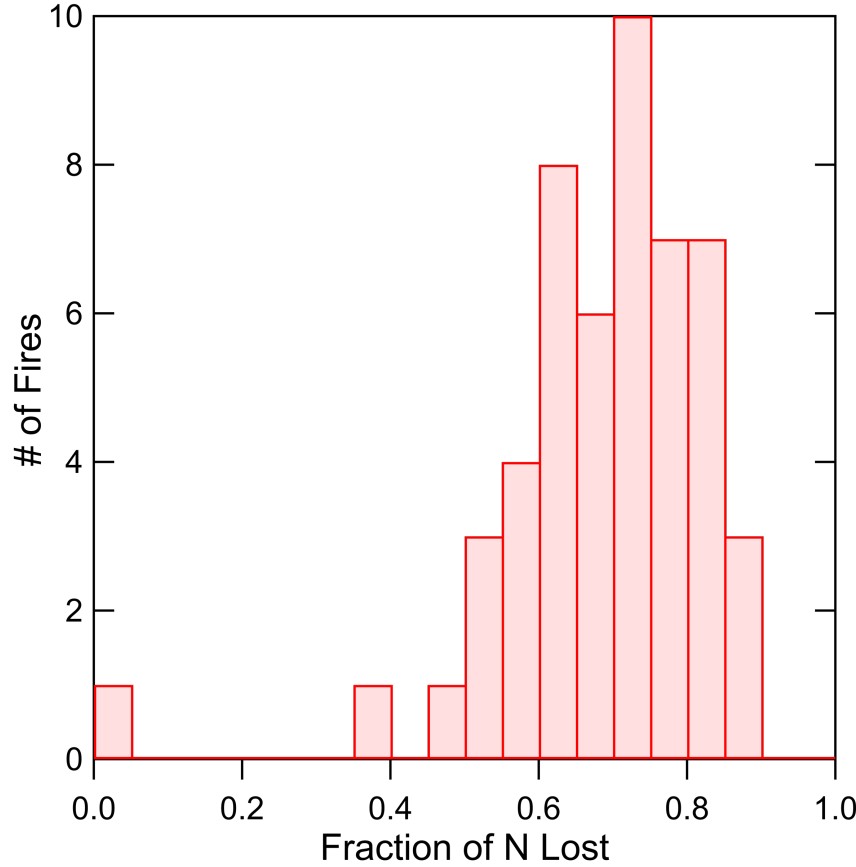

**Figure 4. The histogram of the fraction of N loss to N$_2$ and N$_2$O estimated from the mass**
**balance analysis described in the Supplemental Materials (52 burns).**

**Figure 5. Timelines of $N_r$ and NO (panel a), $N_r$ – NO, the sum of all measures N species except for NO (panel b), residual of $N_r$ minus all measured N species ($N_r$ – NO – Sum N, panel c), and MCE and ($N_r$ – NO)/$N_r$ (panel d) for Fire047, subalpine fir realistic mix.**


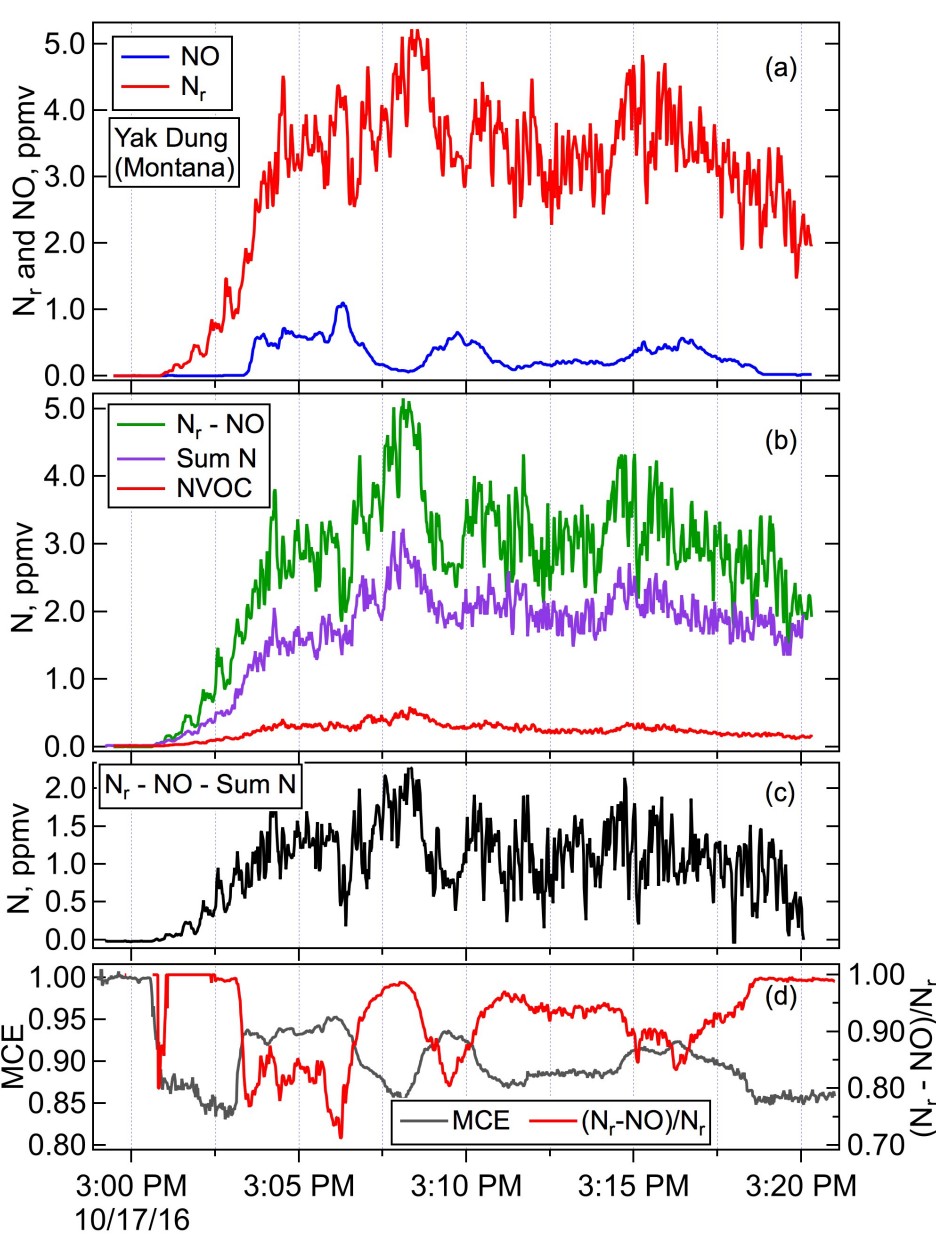

**Figure 6. Timelines of $N_r$ and NO (panel a), $N_r$–NO, the sum of all measured $N_r$ species except**
**for NO (panel b), and residual of $N_r$ minus all measured N species ($N_r$–NO–Sum N, panel c)**
**and MCE and ($N_r$–NO)/$N_r$ (panel d) for Fire050, Montana yak dung.**






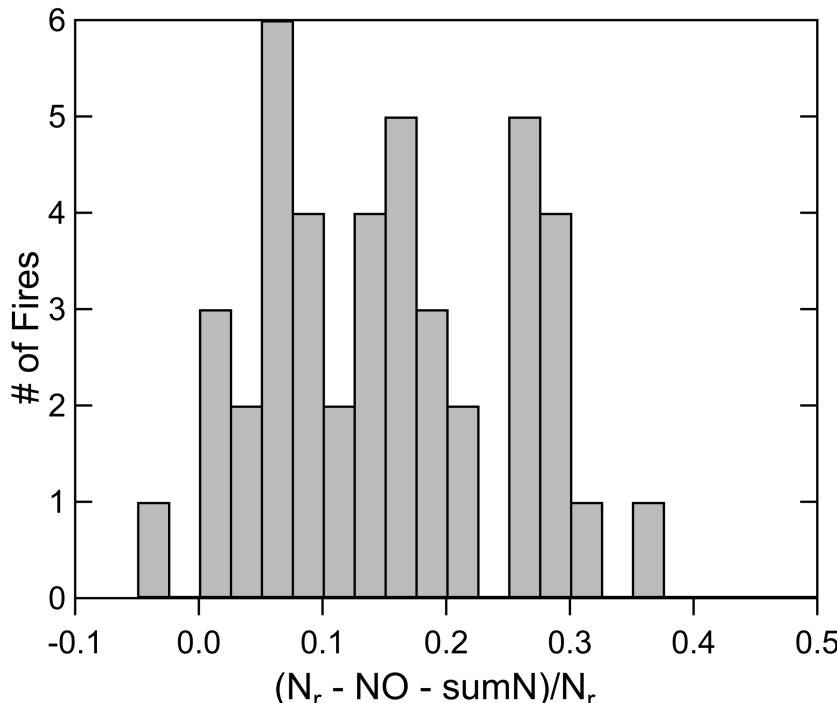

**Figure 7. A histogram of the residual N for all the stack fires during the 2016 FireLab study**
**for which there are FTIR, ACES and PTR-ToF measurements (n=43). The median is 0.143,**
**and the mean (±std dev) was 0.15 (±0.10).**






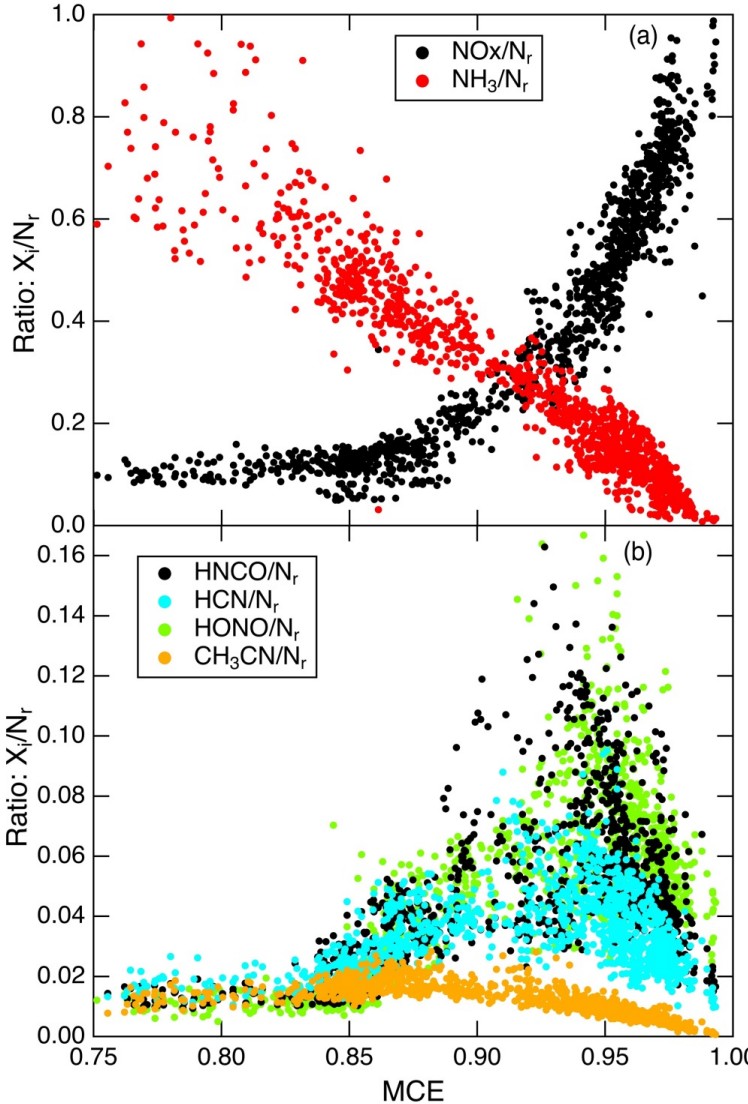

**Figure 8. The relationships between NO$_x$/N$_r$ and NH$_3$/N$_r$ vs MCE (panel a), and the**
**HNCO/N$_r$, HCN/N$_r$, HONO/N$_r$, and CH$_3$CN/N$_r$ vs MCE (panel b) for Fire 047.**




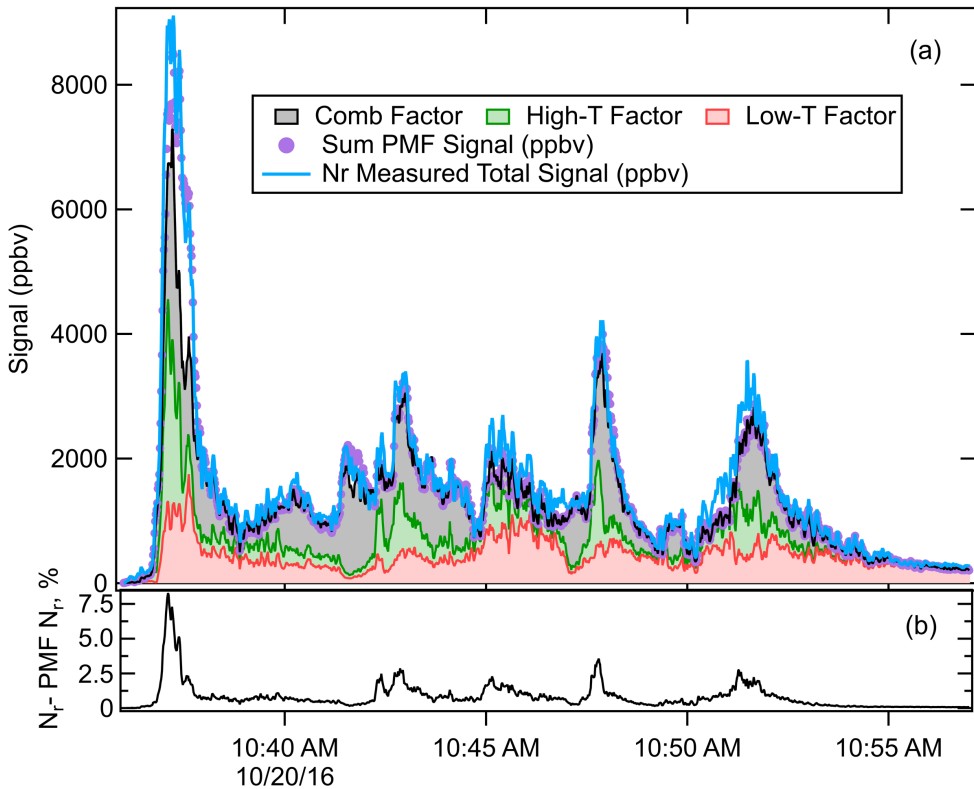

**Figure 9. Pane (a), the measured $N_r$ signal for Fire 063 (lodgepole pine) (blue line), the sum**
**of the signal reconstructed by the PMF (purple points) and the three PMF factors:**
**combustion (grey), high temperature (green) and low temperature (red), plotted in a stacked**
**fashion (i.e. added on top of one another). Panel (b) the "residual" of the PMF fit consisting**
**of the measured $N_r$ signal minus the $N_r$ signal reconstructed by the PMF, as a percentage of**
**the $N_r$ signal.**

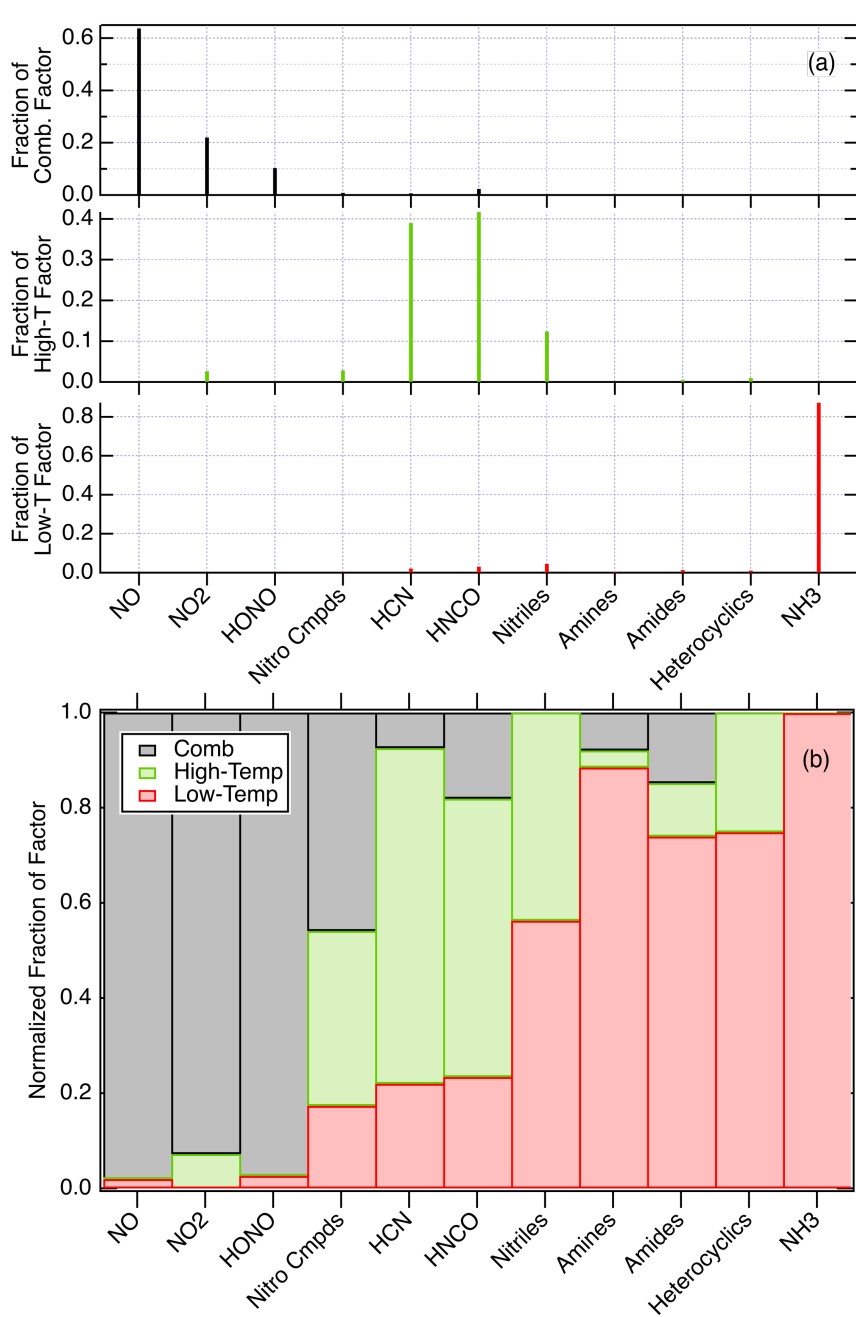

**Figure 10. The contributions of nitrogen species to the factors that simulate the emissions**
**from coniferous fuels shown in Figure S2 (panel a), and the fraction of each compound or**
**class found in each factor (panel b).**



**Figure 11. Comparisons of the N-PMF combustion factor (Comb-N) with CO₂ (Panel a)**
**and MCE (Panel b) for Fire 037 (ponderosa pine). Panel (c) shows the scatter plot of the**
**Comb-N factor versus CO₂ and panel (d) shows the scatter plot of Comb-N factor versus**
**MCE.**



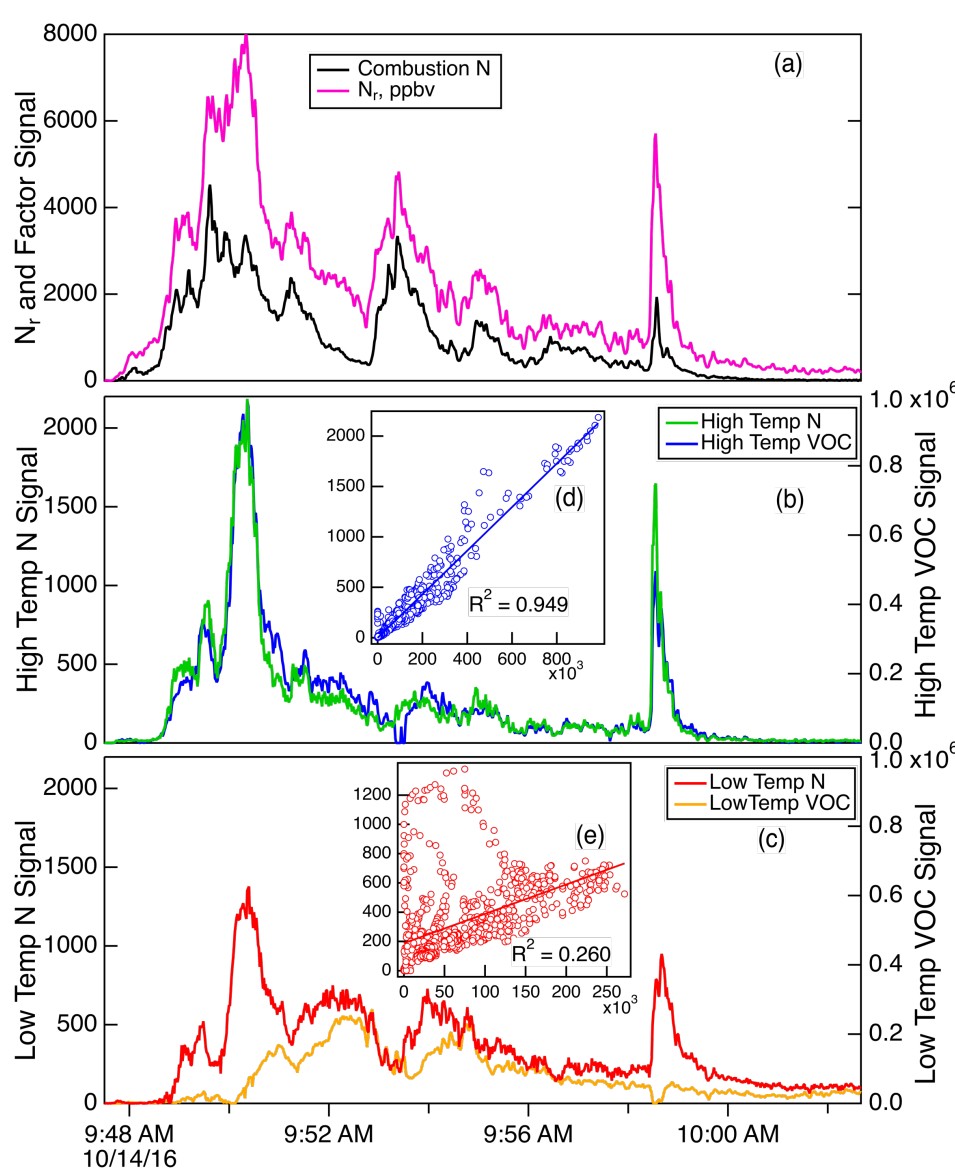

**Figure 12. Details of the PMF factors for Fire 037 (ponderosa pine). Panel (a) shows the total**
**$N_r$ signal (magenta) and the Comb-N factor (black), panel (b) shows the HT-N factor (green)**
**and HT- VOC factor (blue), and panel (c) shows the LT-N factor (red) and LT-VOC factor**
**(orange). The insets (panel d) show the correlation of the two HT factors and the correlation**
**between the two LT factors.**