# Peer review of "The nitrogen budget of laboratory-simulated western U.S. wildfires during the FIREX 2016 FireLab study"

_Atmospheric Chemistry and Physics, 2020_

## Referee Comment (RC1) · Anonymous Referee #1 · 8 Apr 2020

General comments:

The authors present an overview of the N-budget during the FIREX campaign, bringing together datasets from a variety of instruments and techniques (outlines in Table 1) in already peer-reviewed articles of FIREX work. This analysis is certainly a great contribution to the field and specifically to the science of reactive nitrogen budgets from forest fires. The methods used are state-of-the-art, including the instrumentation used for acquiring the data during the campaign in addition to the data analysis, including PMF. I command the authors on putting this data set together and anticipate this contribution, upon revision, to be a go-to paper for the atmospheric organic-N and inorganic-N fate

from biomass burning, well fitting within the scope of Atmos. Chem. Phys.

My most significant revision recommendation is related to the writing and to the presentation of the research. It is difficult to read the manuscript as the information is presented in no specific order and without pertinent subsections. Lots of information is relayed, but it is unclear what the scientists set out to do in the first place. My specific feedback is below to improve the manuscrupt.

Scientific feedback:

It is unclear to me exactly what the conclusion of the work is. Is it that N mass balance was achieved in this study (stated in lines 298-299 – which seems to be a particularly important finding in my opinion, even one to be highlighted as the title, or at least feature predominantly in the abstract)? Is that Nr can/cannot be predicted (mentioned on lines 337-338)? Is it that the dominant molecules of Nr were identified? Is it that an estimated Nr/C ratio of 0.37% can be recommended for modeling (lines 450-452)? It seems like the authors touch upon each one of these questions but the support for these conclusions can be better communicated. In my opinion, they are all important questions for the field which this paper nicely addresses, but currently in a rather disordered fashion. My advice is to use subsections addressing each one of the scientific questions listed (and others as the authors see fit) and present the data in a logical way: (1) state the hypothesis and research question; (2) show the appropriate figure and state the results; (3) discuss the implications of the work.

The Comb-N, HT-N and LT-N factors represent different classes of Nr emissions during biomass burning. What was the big picture goal here? Was it to know that for example when NH3 is detected in a plume, that temperatures related to LT-N are taking place at the origin for the fire? In other words, are these identified chemical markers used to estimate the temperature and burning stage of the fire? If so, then the authors should add this goal to their abstract, highlight this result in their conclusion and devote a titled sub-section to this analysis.

Which research question was being addressed by comparing chaparral fuels within this study? Is the information distracting from the other main messages? Is it necessary to include the analysis related to the chaparral fuels in this paper?

Why place an emphasis on Batch 1 and Batch 2 (including Table 2) if the conclusions are that the same factors were obtained irrespective of which "Bacth was used" (explained in lines 367-368)? Is it necessary to describe both of these PMF analysis? I would argue (although happy to be convinced otherwise) that this description is not necessary for the analysis presented in this manuscript. Only one batch could be described (would also affect Table 2) and a simple one-sentence mention that the same factors were obtained with and without inclusion of CO2, CO and Nr.

I'm curious to know whether there is a "time" component expected for the Nr budget. In other words, would the Nr species evolve over time away from a plume of biomass burning? If so, how? There is mention of flame chemistry on lines 348-362 and Figure 8 (where there is a clear behavior of multiple generation production). For example, the brown carbon properties of biomass burning aerosols respond to heterogeneous oxidation (see (Browne et al., 2019)). For example, are the organic-nitrogen compounds evolving from amines to amides to isocyanates (see (Borduas et al., 2016))?

HNO3 is mentioned in the text (lines 87; 94-98; 300-301) but it wasn't clear where its discussion fits within the topic of Nr. Could the authors specify the significance and consider adding a subsection on HNO3?

Lines 281-289: Is the goal of this section to update MCE values with this work, or to compare the MCE values to the literature? Which values are to be used in further modelling for instance?

I'm curious about the atmospheric implications of yak dung (lines 307-310).

Based on the results from FIREX, would the authors continue to recommend acetonitrile (line 311) as an adequate biomass burning tracer?

[Figure]

Lines 322-323: HNCO and HCN are organic compounds (of which the definition is any molecule which contain at least one carbon and one hydrogen) and shouldn't be included in the "inorganic N" category and discussion.

Presentation feedback:

The best way to improve the manuscript is by structuring the text. The results are interesting but are buried in paragraphs with a lack of order. Every paragraph could have subsections identifying the main message of the results.

In general, the structure of a paragraph starts with a topic sentence about what the paragraph will describe. I think being attentive to that structure could really help bring flow to this manuscript. (For example, one could also avoid starting paragraphs by pointing to a Figure (lines 271-272)).

The scientific research questions could be better identified and articulated. For example, the abstract starts with stating the method, without giving context, the research question and the hypothesis. A short revision here could help the reader situate the study.

Lines 84-87 cites a long list of references. I would argue that this list is not so useful unless each study is briefly described and the main message is communicated.

Lines 164-166: would be great to show this data (in the suppl info)

Lines 208-211: could give an example of the calculation

Table 1 is a great summary of the techniques and compounds included in this study. However, I was a little confused about the meaning of the "references" column. The references from where the data from FIREX was published should be identified separately. I think there are currently 5 publications from FIREX which this study correctly references. What is the meaning of the references prior to 2016? Are they instrument references? It would be great to add further details here as well as provide all the acronyms of the instruments in the caption.

Figure 1: The equations aren't mass/atom balanced. For example, HCN → O, OH → CN is a misleading representation of the reaction. One can use curve arrows to show the intermediates with their own products. Or one can add all the products next to the specie. The chemical equations should be mass balances in any case. In addition, it would also be useful to denote all radical species (either with a dot or another way). I would avoid the (s) notation at it is typically reserved for the solid phase. Perhaps $H_2O$ + surface could be clearer.

Lines 386-396: an important message! So temperature is a better predictor than MCE? Interesting!

Technical comments:

Lines 37-43: This sentence is 7 lines long and could be broken down into 3-4 sentences.

Line 44: define chaparral fuels

Line 76: use arrow instead of "=>"

Lines 115-123: best to use the present tense in this paragraph rather than future tense (which reads more a like a proposal)

Line 256: Is the "emitted carbon" in the gas phase or the particle phase?

Figure 3: the x-axis in 103 ppmv-sec was a little misleading. Took me a few minutes to understand/see that there was a factor of 1000 between the y-axis and the x-axis. Could be worth highlighting these units differently.

Figure 10: top panel is arguably redundant and could be removed. (same for the equivalent figure in the suppl. Infor)

Table 3: Could be better represented as a graph?

References:

Borduas, N., Abbatt, J. P. D., Murphy, J. G., So, S. and da Silva, G.: Gas-Phase Mechanisms of the Reactions of Reduced Organic Nitrogen Compounds with OH Radicals, Environ. Sci. Technol., 50(21), 11723–11734, doi:10.1021/acs.est.6b03797, 2016.

Browne, E. C., Zhang, X., Franklin, J. P., Ridley, K. J., Kirchstetter, T. W., Wilson, K. R., Cappa, C. D. and Kroll, J. H.: Effect of heterogeneous oxidative aging on light absorption by biomass burning organic aerosol, Aerosol Sci. Technol., 0(0), 1–12, doi:10.1080/02786826.2019.1599321, 2019.

---

## Referee Comment (RC2) · Anonymous Referee #2 · 13 Apr 2020

The paper presents a unique and high quality set of observations of reactive nitrogen from smoke emanating from laboratory burns of predominantly western U.S. fuels. Variability in the emissions of specific reactive nitrogen species are modeled using a positive matrix factorization (PMF), and the paper makes recommendations on specific markers to be used for emissions of reactive nitrogen emissions from combustion, high-temperature pyrolysis and low-temperature pyrolysis. The overall content is already largely suitable for publication in ACP, but some improvements to the structure could make it much easier to digest. Thus most of my comments are editorial in nature.

Edits:

[Figure]

Line 135: It seems like particle phase measurements were made during the FIREX burns? I immediately wondered. ...Why is there no use of the LTOFAMS data to compare to the "particle-bound" species that are "not included in this analysis"?

Section 2.3 PMF Analysis: The methods/details in this section really do need to be expanded so that this analysis is actually reproducible. Please add Q/Qexp values, FPEAK values, and the number of bootstrapping runs for all calculations.

This section would substantially benefit from some better organization. I had to read many of the paragraphs twice to make sure I understood them and I often felt like the order was random. I recommend looking through this section and dividing it into several new more specific sub-sections, rather than just one sub-section (i.e. current 3.1). Perhaps it would be better to have non-western U.S. fuels (e.g. the Yak Dung) just appear in the SI, rather than in the main text. This might help Section 3 feel more focused.

Comments Specific to Figures:

Figure 1: This is really nice. Could you add a list (or denote in some way – that would be even better) all the species not measured in this study?

I think there is value in having Figure 2 in the main text. It is nice because it shows the evolution of the fire, and how the reactive nitrogen and carbon-containing species evolve as a result. However, the use of Figure 5 and Figure 6 feels tedious. The text is sufficiently wordy that the reader has to go into those time series and try to interpret/summarize the patterns themselves. I would recommend that there is only one time series Figure and then the others are moved to the SI. If there is something specific to see in Figure 5 and Figure 6 that is contrasting between the fires or called-out in the text, then those sections of the plots should be highlighted somehow, maybe with transparent yellow bars.

Figure 3: Why is Duff twice without noting differences between them?

**[ACPD](https://doi.org)**

Interactive
comment

Figure 11: I would combine the top two panels of Figure 11. Why show a R2 of a linear fit in panel d) when that relationship is not linear?

Minor Edits:

Line 97: Combine parentheses around citations.

Lines 115 – 123: This paragraph should be in present tense, not future. The jump to future tense here is disorienting.

Line 168: change "into" to "by"

---

## Author Comment (AC1) · 2 Jun 2020

Enclosed are responses to reviewers and annotated main and supplementary text. Note that Table S1, an Excel file, has not changed.

Please also note the supplement to this comment: https://www.atmos-chem-phys-discuss.net/acp-2020-66/acp-2020-66-AC1-supplement.zip

---

## Author Response (AR1)

*Our responses are interspersed with the reviewers' comments and are given in red italic text. The line numbers where those changes appear in the revised paper are also given at that point. The revised main text and supplemental information are posted as additional author comments.*

Anonymous Referee #1
General comments:
The authors present an overview of the N-budget during the FIREX campaign, bringing together datasets from a variety of instruments and techniques (outlines in Table 1) in already peer-reviewed articles of FIREX work. This analysis is certainly a great contribution to the field and specifically to the science of reactive nitrogen budgets from forest fires. The methods used are state-of-the-art, including the instrumentation used for acquiring the data during the campaign in addition to the data analysis, including PMF. I command the authors on putting this data set together and anticipate this contribution, upon revision, to be a go-to paper for the atmospheric organic-N and inorganic-N fate from biomass burning, well fitting within the scope of Atmos. Chem. Phys.
*We thank the reviewer for their kind words and hope that our revisions will meet with approval.*

My most significant revision recommendation is related to the writing and to the presentation of the research. It is difficult to read the manuscript as the information is presented in no specific order and without pertinent subsections. Lots of information is relayed, but it is unclear what the scientists set out to do in the first place. My specific feedback is below to improve the manuscrupt.
*We have revised the manuscript to better organize and emphasize the various subsections. Specific changes involved dividing the Results Discussion section into subsections and adding introductory sentences describing the specific research question to be answered. Never-the-less, we do believe the information is presented in the correct order, as the Results section proceeds from the most general (total $N_r$ and total carbon) to the most specific (factors that define emissions composition). We hope that the division into sub-sections, with the addition of introductory material will make that more apparent. We think the last sentence of the Introduction adequately explains what we set out to do in the first place "The results are used to arrive at suggested guidelines that can be used estimate $N_r$-emissions profiles for fires representative of western North America."*

Scientific feedback:
It is unclear to me exactly what the conclusion of the work is. Is it that N mass balance was achieved in this study (stated in lines 298-299 – which seems to be a particularly important finding in my opinion, even one to be highlighted as the title, or at least feature predominantly in the abstract)? Is that $N_r$ can/cannot be predicted (mentioned on lines 337-338)? Is it that the dominant molecules of $N_r$ were identified? Is it that an estimated $N_r$/C ratio of 0.37% can be recommended for modeling (lines 450-452)?
*Yes, the reviewer has correctly listed most of the major findings of this work. We would add that we show how the composition of $N_r$ has a systematic dependence on the temperature of combustion/reaction in answer to the reviewers' question RE lines 337-338. We have added a sentence about the N/C ratio to the abstract to better emphasize that conclusion. We have also reformulated the budget statement in the abstract to say that measured individual $N_r$ species accounted for 84.8 (±9.8)% of the measured $N_r$ on average. We point out that all of these findings appear in the section labeled "Conclusions". Never-the-less, we have taken steps to better emphasize them as we have reorganized the paper as outlined below.*

It seems like the authors touch upon each one of these questions but the support for these conclusions can be better communicated. In my opinion, they are all important questions for the field which this paper nicely addresses, but currently in a rather disordered fashion. My advice is to use subsections addressing each one of the scientific questions listed (and others as the authors see fit) and present the data in a logical way: (1) state the hypothesis and research question; (2) show the appropriate figure and state the results; (3) discuss the implications of the work.

*We thank the reviewer for their advice. We have reorganized the paper, including more subsections and expository statements to guide the reader, as the reviewer has suggested. Specifics changes are shown in red in the Revised paper.*

The Comb-N, HT-N and LT-N factors represent different classes of $N_r$ emissions during biomass burning. What was the big picture goal here? Was it to know that for example when $NH_3$ is detected in a plume, that temperatures related to LT-N are taking place at the origin for the fire? In other words, are these identified chemical markers used to estimate the temperature and burning stage of the fire? If so, then the authors should add this goal to their abstract, highlight this result in their conclusion and devote a titled sub-section to this analysis.

*The big-picture goal here was to look for relationships between the detailed N emissions that we measured and simple measures or marker compounds that could serve as a means to estimate emissions when detailed measurements were not available. We started with MCE, because of the extensive history that MCE has with the wildfire emissions community, and found that MCE does not correlate (either positively or negatively) with an important fraction of $N_r$. As a result, we used PMF to find that three factors were much better able to describe WF emissions and their variation corresponded to roughly three temperature regimes. These temperature regimes made sense in relation to what we know about combustion and pyrolysis chemistry. We then could select several chemical markers to help in representing these combustion regimes. We have now added emphasis to the abstract and the re-organization of the Results and Discussion section provides the sub-sections requested by the reviewer. We have further emphasized this result in the Conclusions.*

Which research question was being addressed by comparing chaparral fuels within this study? Is the information distracting from the other main messages? Is it necessary to include the analysis related to the chaparral fuels in this paper?

*The intent of this phase of the FIREX project was to study emissions from fuels characteristic of Western North American Wildfires. Chaparral ecosystems are important throughout Central and Southern California and other areas of the Southwestern U.S. Thus, we feel the research on chaparral fuels belongs in this paper. We mention the importance of chaparral fuels in the paper on Lines 142-143.*

Why place an emphasis on Batch 1 and Batch 2 (including Table 2) if the conclusions are that the same factors were obtained irrespective of which "Bacth was used" (explained in lines 367-368)? Is it necessary to describe both of these PMF analysis? I would argue (although happy to be convinced otherwise) that this description is not necessary for the analysis presented in this manuscript. Only one batch could be described (would also affect Table 2) and a simple one-sentence mention that the same factors were obtained with and without inclusion of CO2, CO and Nr.

*We agree and have now changed the discussion to note the results were the same when including CO, $CO_2$ and $N_r$, and present only the runs previously denoted 'Batch 2" in Table 2 and results. Lines 237-240.*

I'm curious to know whether there is a "time" component expected for the Nr budget. In other words, would the Nr species evolve over time away from a plume of biomass burning? If so, how? There is mention of flame chemistry on lines 348-362 and Figure 8 (where there is a clear behavior of multiple generation production). For example, the brown carbon properties of biomass burning aerosols respond to heterogeneous oxidation (see (Browne et al., 2019)). For example, are the organic-nitrogen compounds evolving from amines to amides to isocyanates (see (Borduas et al., 2016))?

*A goal of this work was to investigate how combustion partitioned fuel N. In these experiments in the Firelab the emissions undergo essentially no further processing once they leave the flame /smoldering zone. So, no there is not a "time" component to the $N_r$ budget in the sense that the reviewer means, i.e. based on the quoted references. As a result, discussion of these aspects of processing is not appropriate*

*for this paper, which focusses on emissions. Yes, there is a whole rich body of work on BB atmospheric processing, some of which was done as part of this FIREX FireLab study, but it does not apply to our topic.*

HNO3 is mentioned in the text (lines 87; 94-98; 300-301) but it wasn't clear where its discussion fits within the topic of Nr. Could the authors specify the significance and consider adding a subsection on HNO3?

*$HNO_{3(g)}$ would be easily measured by the $N_r$ and OP-FTIR technique if it was present, and therefore needs to be mentioned. What we perhaps did not make clear is that $HNO_3$ is not observed above detection limit (10 ppbv). This was due to the large particle surface area loading and the high $NH_3$ present, both favoring immediate conversion of any $HNO_{3(g)}$ to particle nitrate, essentially by the time the smoke reached the top of the stack (5 sec). This does not warrant another section, instead we have now made this clearer in the text and in the discussion: Lines 161-163, and Table 1.*

Lines 281-289: Is the goal of this section to update MCE values with this work, or to compare the MCE values to the literature? Which values are to be used in further modelling for instance?

*The whole fire MCE values are summarized in Selimovic et al., 2018. The goal of this section of our paper is to contrast our findings on fuel N conversion to $N_2$ and $N_2O$ with previous measurements and estimates in the literature. Since high temperature combustion conditions and associated flame chemistry are required for this conversion, we use this opportunity to place our observations in the context of the previous studies by using MCE as the key indicator of this combustion chemistry. We do not use MCE in any modeling as we later show, it does not capture the dependence of emissions on temperature.*

I'm curious about the atmospheric implications of yak dung (lines 307-310).

*Yak dung is an important domestic fuel in a number of developing countries, especially in ecosystems above timberline or where wood is scarce. These emissions can impact outdoor air chemistry in areas such as India, Nepal, and Tibet, (see Xiao et al., 2015 for example). In addition, Yak dung emissions have not been measured by a suite of instruments as extensive as those used in this work, and represent something of an extreme case of a high nitrogen/carbon fuel that typically burns at lower MCEs. Lines 143-144.*

Based on the results from FIREX, would the authors continue to recommend acetonitrile (line 311) as an adequate biomass burning tracer?

*In many cases, acetonitrile is a very good tracer of biomass burning, but Coggon et al., (2016) noted that solvent sources of acetonitrile can sometimes obscure the BB signature, and that acetonitrile is sometimes quite low in emissions from heartwood that is often burned in woodstoves. HCN is a more reliable signature for the HT-N pyrolysis stage of WF emissions and it is also high in peat and dung fires so can serve as a good tracer for those too. We have now noted this in lines 477-480.*

Lines 322-323: HNCO and HCN are organic compounds (of which the definition is any molecule which contain at least one carbon and one hydrogen) and shouldn't be included in the "inorganic N" category and discussion.

*This is a very interesting area of discussion and could take up a lot of time if we tried to cover it in this paper. By the Reviewer's definition, sodium bicarbonate ($NaHCO_3$) and carbonic acid ($H_2CO_3$) would be considered organic compounds, but the vast majority of scientists would disagree with that. More complete definitions specify that an organic compound contains carbon, hydrogen, and perhaps oxygen, nitrogen and sulfur, **covalently bound**. This is the key to classifying HNCO and HCN, as these H-C and H-N bonds are ionic, not covalent: both compounds are weak acids, and dissociate in aqueous solution.*

Presentation feedback:

The best way to improve the manuscript is by structuring the text. The results are interesting but are buried in paragraphs with a lack of order. Every paragraph could have subsections identifying the main message of the results.

In general, the structure of a paragraph starts with a topic sentence about what the paragraph will describe. I think being attentive to that structure could really help bring flow to this manuscript. (For example, one could also avoid starting paragraphs by pointing to a Figure (lines 271-272)).
*As described above, we now have divided the Results and Discussion section into more subsections, with the addition of introductory sentences where needed, so that we are not starting sub-sections by pointing to a Figure. We note that occasionally it makes sense to start paragraphs within a sub-section by introducing a new or contrasting figure that is part of the larger topic of that sub-section.*

The scientific research questions could be better identified and articulated. For example, the abstract starts with stating the method, without giving context, the research question and the hypothesis. A short revision here could help the reader situate the study.
*We have added sentences to the beginning of the abstract to accomplish this.*

Lines 84-87 cites a long list of references. I would argue that this list is not so useful unless each study is briefly described and the main message is communicated.
*We have now provided a brief description for those papers on Lines 91-97.*

Lines 164-166: would be great to show this data (in the suppl info).
*A discussion of the effect of diffusion and dispersion due to laminar flow is now included in the SI, along with comparison of the NO and $N_r$ signals to those of the OP-FTIR, that show the effective 4 sec time constant. And we note that in Lines 186-187*

Lines 208-211: could give an example of the calculation
*We now give an example calculation at this point. Line 231-232.*

Table 1 is a great summary of the techniques and compounds included in this study. However, I was a little confused about the meaning of the "references" column. The references from where the data from FIREX was published should be identified separately. I think there are currently 5 publications from FIREX which this study correctly references. What is the meaning of the references prior to 2016? Are they instrument references? It would be great to add further details here as well as provide all the acronyms of the instruments in the caption.
*The 'References' column is now relabeled 'Method Reference' to make it clear that these are the publications that described the methods. As a result, several pre-date this lab study. We have spelled out the acronyms, as requested.*

Figure 1: The equations aren't mass/atom balanced. For example, HCN → O, OH → CN is a misleading representation of the reaction. One can use curve arrows to show the intermediates with their own products. Or one can add all the products next to the specie. The chemical equations should be mass balances in any case. In addition, it would also be useful to denote all radical species (either with a dot or another way). I would avoid the (s) notation at it is typically reserved for the solid phase. Perhaps $H_2O$ + surface could be clearer.
*Indeed, there are reactions presented here that are not balanced. The purpose of presenting this material in Figure form is to make it easy to see the general flow of the chemistry with a rough separation by temperature regime. A more thorough representation of the chemistry with the various mechanisms and products would unnecessarily clutter a diagram like Figure 1. What the reviewer seems to be asking for is an exhaustive listing of balanced equations, which is really beyond the goals of this paper, and in fact is*

*very ably covered by Glarborg et al., (2018). We have added more details across the bottom of the diagram, so for example $CN + O_2 \rightarrow NCO + O$ now form a balanced equation. We have added dots to denote radical species as requested. We now use $H_2O_{(surf)}$ to denote surface-adsorbed water. We note that Reviewer 2 liked Figure 1, and requested only a few additions as noted below. We have added a sentence to clarify the intent of Figure 1, (Lines 75-77).*

Lines 386-396: an important message! So temperature is a better predictor than MCE? Interesting!
*The word "predictor" is difficult because we don't have a measure of the actual fire temperature. We prefer to think of it as the $N_r$ speciation correlating with temperature as indicated by key chemical species, and yes, we feel this is one of the key conclusions of the work.*

Technical comments:

Lines 37-43: This sentence is 7 lines long and could be broken down into 3-4 sentences.
*We have removed a redundant clause and changed the last clause into a sentence.*

Line 44: define chaparral fuels
*We have specified manzanita and chamise on Line 48.*

Line 76: use arrow instead of "=>"
*Done, and an arrow was also added to the chemical equation that was on line 77. Lines 83-84.*

Lines 115-123: best to use the present tense in this paragraph rather than future tense (which reads more a like a proposal).
*By convention, the last paragraph in an Introduction describes what the paper will present, rather what has already been done in the area of research being addressed. As a result, phrasing this paragraph in the future tense could help to make this apparent. However, we acknowledge the use of present tense is common and we have changed to the present tense since both reviewers appeared to take issue with this approach.*

Line 256: Is the "emitted carbon" in the gas phase or the particle phase?
*It is both, which seems clear from the sentence "The additional carbon species included methane and a number of other gas phase VOCs as well as organic- and black-carbon aerosol."*

Figure 3: the x-axis in 103 ppmv-sec was a little misleading. Took me a few minutes to understand/see that there was a factor of 1000 between the y-axis and the x-axis. Could be worth highlighting these units differently.
*We now highlight the different in scales in the figure caption. We think this, and the fact that the figure has lines ranging from 0.25% to 2% drawn on it should make the scale difference clear.*

Figure 10: top panel is arguably redundant and could be removed. (same for the equivalent figure in the suppl. Infor)
*We disagree. The top panels in these figures show different information from the bottom panels of these figures. The top panels show the fraction of each factor that is accounted for by each of the compounds or subclasses, while the bottom panel shows how the compounds or subclasses are distributed among the three factors, so by nature, each compound or class in Figure 10b adds up to 1.0. Take for example the subclass 'nitro-compounds', Figure 10b shows how it is divided among the three factors, but Figure 10a shows that it is a relatively small contributor to any one factor. There is no way to get that information from Figure 10b. These features of Figure 10 were clearly stated in the text (Lines 431-433). Note that Figure 10 in the original is now Figure 9.*

Table 3: Could be better represented as a graph?
*We have considered this and wish to keep this as a Table. The reason being that many current models are not equipped to use the temperature-dependent emissions information we are presenting, but can include simple representations of N emissions. In those cases, these tabular data are most convenient.*

* * *
Anonymous Referee #2
The paper presents a unique and high quality set of observations of reactive nitrogen from smoke emanating from laboratory burns of predominantly western U.S. fuels. Variability in the emissions of specific reactive nitrogen species are modeled using a positive matrix factorization (PMF), and the paper makes recommendations on specific markers to be used for emissions of reactive nitrogen emissions from combustion, high-temperature pyrolysis and low-temperature pyrolysis. The overall content is already largely suitable for publication in ACP, but some improvements to the structure could make it much easier to digest. Thus most of my comments are editorial in nature.

Edits:

Line 135: It seems like particle phase measurements were made during the FIREX burns? I immediately wondered… Why is there no use of the LTOFAMS data to compare to the "particle-bound" species that are "not included in this analysis"?
*The particle measurements at the top of the stack were limited to the optical measurements described by Selimovic et al. (2018), particle organic species measurements described by Jen et al., (2019), and a Compact Time-of-Flight Aerosol Mass Spectrometer (C-ToF-AMS). The CToF-AMS had only unit-mass resolution and because of that, and large deviations from typical mass fragmentation patterns, could not give quantitative measures of ammonium or inorganic nitrate. There were HiResToF and LToF-AMS instruments used for the chamber experiments associated with the FIREX-Firelab project but the aerosol was often manipulated in novel ways to better understand aerosol properties, and processing, and that precluded measuring emission factors. Hence, while they were able to look at process chemistry, they did not sample direct emissions.*

Section 2.3 PMF Analysis: The methods/details in this section really do need to be expanded so that this analysis is actually reproducible. Please add Q/Qexp values, FPEAK values, and the number of bootstrapping runs for all calculations.
*We have now described those in Sections 3.5 and 3.7, and we have added information on our original PMF of Fire 063 in the SI that shows the robustness of the analysis for Fpeak =-1, 0, and +1, and bootstrapping runs with 100 different seeds. Lines 423-427.*

This section would substantially benefit from some better organization. I had to read many of the paragraphs twice to make sure I understood them and I often felt like the order was random. I recommend looking through this section and dividing it into several new more specific sub-sections, rather than just one sub-section (i.e. current 3.1). Perhaps it would be better to have non-western U.S. fuels (e.g. the Yak Dung) just appear in the SI, rather than in the main text. This might help Section 3 feel more focused.

*We have reorganized this section substantially due to both this Reviewers' and Reviewer #1's comments. This reorganization involved dividing Section 3 into a number of smaller subsections, and moving the original Figure 6 (yak dung) to the SI.*

Comments Specific to Figures:

Figure 1: This is really nice. Could you add a list (or denote in some way – that would be even better) all the species not measured in this study?

*We now note that species measured in this work are shown in bold and slightly larger font, and the species not measured are shown in normal text.*

I think there is value in having Figure 2 in the main text. It is nice because it shows the evolution of the fire, and how the reactive nitrogen and carbon-containing species evolve as a result. However, the use of Figure 5 and Figure 6 feels tedious. The text is sufficiently wordy that the reader has to go into those time series and try to interpret/summarize the patterns themselves. I would recommend that there is only one time series Figure and then the others are moved to the SI. If there is something specific to see in Figure 5 and Figure 6 that is contrasting between the fires or called out in the text, then those sections of the plots should be highlighted somehow, maybe with transparent yellow bars.

*We have now moved Figure 6 into the SI. We wish to retain Figure 5 because it shows a number of things the Figure 2 does not, specifically the details of the $N_r$ composition, and that Residual $N_r$ evolved with time during the fire. That the Residual $N_r$ is relatively higher at the end of the fire when smoldering emissions are more prevalent, is now shown with a yellow box in Figure 5.*

Figure 3: Why is Duff twice without noting differences between them?

*We have now labeled the different types of duff on Figure 3b.*

Figure 11: I would combine the top two panels of Figure 11. Why show a R2 of a linear fit in panel d) when that relationship is not linear?

*We have now combined the top two panels into one that shows, $CO_2$, MCE, and Comb-N Factor. The $R^2$ for (now) panel (c) is given as a means to quantify the deviation of the relationship from linear. Please note that Figure 11 in the original is now Figure 10.*

Minor Edits:

Line 97: Combine parentheses around citations.
*Done.*

Lines 115 – 123: This paragraph should be in present tense, not future. The jump to future tense here is disorienting.
*Changed as noted above.*

Line 168: change "into" to "by"
*Done.*

[revised manuscript text omitted]

**Supplemental Material**

**1. Determination of the diffusion and dispersion time constants of the NO and $N_r$ measurements**

The effect that diffusion and dispersion had on the effective time constants of the NO and $N_r$ measurements was estimated from well understood transport and diffusion equations, and was determined experimentally from comparison of these measurements to the NO and $NH_3$ measurements made by the OP-FTIR at the sampling point at the top of the stack. The effects that sampling the atmosphere into a long tube has on the integrity of temporal information has been discussed by Karion et al., (2010). The figure of merit here is the root-mean square of the distance a gas molecule travels during the time the air sample transits from the stack to the instrument. This distance, $X_{rms}$ can be calculated;

$$X_{rms} = (2Dt)^{1/2} \qquad \text{Eq S1.}$$

where $D$ is the bimolecular diffusion coefficient of the analyte in air, and $t$ is the time the diffusion is allowed to happen, in this case the 14 sec transit time from the stack to the instrument. Longitudinal mixing due to laminar flow, sometimes called dispersion, can add to the effective diffusion and can be estimated using the following relationship:

$$D_{eff} = D + a^2V^2/48D \qquad \text{Eq S2.}$$

where $a$ is the inner radius of the tube, and $V$ is the average flow velocity. Substituting the values of $a$ and $V$, and using $D_{NO} = 0.23$ cm$^2$/sec (Tang et al., 2014), results in $X_{rms} \cong 40$cm. The linear velocity of the gas within the tube is 120 cm/sec, so based on simple diffusion and laminar flow dispersion, the effective time constant of the data acquired at 1 Hz would be degraded to about 2 second or so.

The comparison of the 1 Hz NO and $N_r$ data with NO and $NH_3$ measurements acquired at the top of the stack by the OP-FTIR provides a useful means to check the effective time constant.

Figure S1 shows the comparison of the chemiluminescence instrument measurements with the NO and $NH_3$ OP-FTIR measurements. The OP-FTIR had an effective sample acquisition/averaging time of 1.26 sec, so a 3-point smoothing of OP-FTIR signals results in an effective time constant of approximately 4 seconds.

[Figure]

Figure S1. Comparison of the NO chemiluminescence measurement with that of the OP-FTIR (Panel a) for Fire 057, and comparison of the $N_r$ measurement at the end of Fire 047 with the $NH_3$ measurement from the OP-FTIR (Panel b). Both the OP-FTIR 1.26 sec and the OP-FTIR data smoothed with a 3-pt box car method are shown.

2. Estimating the N lost to $N_2$ and $N_2O$.

The loss of N to $N_2$ and $N_2O$ in the stack fires was estimated using the fuels data compiled in the Supplemental Material of Selimovic et al., [2018], and the ash data listed in

Table S1 of this Supplemental. The fuels data used in the analysis include: Total Fuel Mass, Total Residual Mass, %N Fuel (by weight), %C Fuel (by weight), and the ash data used in the analysis include: the ratio Ash/Burned Fuel, %N Ash (by weight), %Total C Ash (by weight). The gas phase measurements used for the analysis include: Total Reactive Nitrogen ($N_r$) reported here, and Total Carbon ($CO_2$ + CO + $CH_4$ +ΣNMOC + Particle Carbon), calculated in the manner described by Selimovic et al., [2018]. In the calculations below, we assume the Ash/Burned Fuel, %N Ash and %C Ash are the same for the stack burns as the quantities measured during the room burns. These assumptions add only a modest level of uncertainty since the fuels burned in each set of experiments were subsets of large samples of each fuel type, and the use of Ash/burned fuel removes some of the variability in fire conditions and extent, as it accounts for unburned residual fuel. Another source of uncertainty is the application of fuel moisture measurements. In general, the correction for fuel moisture applies equally to foliage (needles) and woody biomass, but there are occasions where those were not equal or the residual fuel was more heavily represented by woody biomass. Residual masses were often 10% of the initial fuel mass, but sometimes as high as 50% of initial mass. Considering these factors, we estimate an uncertainty in the mass balance calculations to be ±25%.

The mass balance equations for Nitrogen and Carbon are;

Mass N emitted = Mass Total Fuel *%N Fuel – Mass Ash*%N Ash – Mass Unburnt residual *%N Fuel = Mass ($N_r$ + $N_2$ + $N_2O$)                              Eq. S3

Where: Mass Unburnt residual = Mass Total Residual – Mass Ash                  Eq. S4.

and

Mass C emitted = Mass Total Fuel *%C Fuel – Mass Ash*%C Ash – Mass Unburnt residual *%C Fuel = Mass ($CO_2$ + CO +$CH_4$ +NMOC + Particle C) = Mass Total C                  Eq. S5

Where: Mass Unburnt residual = Mass Total Residual – Mass Ash                  Eq. S6.

   There are measured concentrations of $N_r$, and Total C, however there were not accurate measurements of the actual flow rates of air up the stack. The concentrations (mixing ratios) of N and C species are related to mass flow by several constants, e.g. pressure, temperature, Avogadro's number, all of which are the same for both N and C, except for the atom weights. As a consequence, we can use the ratios of concentrations to obtain the following relationships;

$\underline{(N_r+N_2+N_2O)}$ = $\underline{\text{(Mass Total Fuel *%N Fuel – Mass Ash*%N Ash – Mass Unburnt residual *%N Fuel)/14g}}$  Eq.S7
  Total C  (Mass Total Fuel *%C Fuel – Mass Ash*%C Ash – Mass Unburnt residual *%C Fuel)/12g

Recognizing that;

$\underline{(N_2+N_2O)}$ =$\underline{\text{(Mass Total Fuel *%N Fuel – Mass Ash*%N Ash – Mass Unburnt residual *%N Fuel)/14g}}$ _ $\underline{N_r}$  Eq.S8
 Total C (Mass Total Fuel *%C Fuel – Mass Ash*%C Ash – Mass Unburnt residual *%C Fuel)/12g Total C

The ratio of Eqs. S8 and S7 gives; ($N_2$+$N_2O$) /($N_r$+$N_2$+$N_2O$) = The fraction of N lost as $N_2$ and $N_2O$, estimated from fuel and ash composition and the measured quantity $N_r$/Total Carbon.

[Figure]

Figure S2. Timelines of $N_r$ and NO (panel a), $N_r$-NO, the sum of all measured $N_r$ species except for NO (panel b), and residual of $N_r$ minus all measured N species ($N_r$-NO-Sum N, panel c) and MCE and ($N_r$-NO)/$N_r$ (panel d) for Fire050, Montana yak dung.

[Figure]

Figure S3. The relative amounts of residual $N_r$ vs MCE (a) and vs $(N_r-sumN)/N_r$ (b) for whole fires. The lines are orthogonal-distance-regression fits that assume uncertainty in each variable.

[Figure]

Figure S4. Details of the PMF analysis of Fire 063. Figures S4 a, c and e show that the Fpeak=0 solution is stable compared to Fpeak -1 and +1. Figures b, d, and f show that the three factor solution is robust with respect to different initial factor profiles (seeds) for 100 different runs.

[Figure]

Figure S5. Combined PMF timeline for the fires that involved coniferous fuels. The measured $N_r$ is shown as a blue line, the total of $N_r$ compounds used in the PMF is shown as purple points, and Comb-N (grey), HT-N (green), and LT-N (red) factors plotted stacked on top of one another. The vertical lines show where individual fires start and stop.

[Figure]

Figure S6. The correlation of HONO/NOx (by mole) with needle moisture for fires that were canopy fuels only (Fires 015, 017, 018, 019, 020, 023, 025, 039, 040, 044, 045, and 064).

[Figure]

Figure S7. The timeline for the combined PMF analysis of chaparral fuels. The measured $N_r$ is shown as a blue line, the total of N compounds used in the PMF is shown as purple points, and Comb-N (grey), HT-N (green), and LT-N (red) factors plotted stacked on top of one another. The vertical lines show where individual fires start and stop.

[Figure]

Figure S8. The contributions of nitrogen species to the factors that simulate the emissions from chaparral fuels shown in Figure S7 (panel a), and the fraction of each compound or class found in each factor (panel b).